# Extensive androgen receptor enhancer heterogeneity in primary prostate cancers underlies transcriptional diversity and metastatic potential

Jeroen Kneppers [1], Tesa M. Severson[1,2], Joseph C. Siefert[1,2], Pieter Schol[1], Stacey E. P. Joosten[1], Ivan Pak Lok Yu[3], Chia-Chi Flora Huang [3], Tunç Morova[3], Umut Berkay Altıntaş [4], Claudia Giambartolomei[5,6], Ji-Heui Seo[6,7], Sylvan C. Baca[7], Isa Carneiro[8], Eldon Emberly[9], Bogdan Pasaniuc[6], Carmen Jerónimo[8], Rui Henrique [8], Matthew L. Freedman [7,10], Lodewyk F. A. Wessels [2], Nathan A. Lack[3,4,11], Andries M. Bergman [1,12] ✉ & Wilbert Zwart [1,13] ✉

Androgen receptor (AR) drives prostate cancer (PCa) development and progression. AR chromatin binding profiles are highly plastic and form recurrent programmatic changes that differentiate disease stages, subtypes and patient outcomes. While prior studies focused on concordance between patient subgroups, inter-tumor heterogeneity of AR enhancer selectivity remains unexplored. Here we report high levels of AR chromatin binding heterogeneity in human primary prostate tumors, that overlap with heterogeneity observed in healthy prostate epithelium. Such heterogeneity has functional consequences, as somatic mutations converge on commonly-shared AR sites in primary over metastatic tissues. In contrast, less-frequently shared AR sites associate strongly with AR-driven gene expression, while such heterogeneous AR enhancer usage also distinguishes patients' outcome. These findings indicate that epigenetic heterogeneity in primary disease is directly informative for risk of biochemical relapse. Cumulatively, our results illustrate a high level of AR enhancer heterogeneity in primary PCa driving differential expression and clinical impact.

## Androgen receptor usage is heterogeneous in primary prostate cancer patients

Prostate cancer (PCa) has the second highest incidence in men worldwide[1] and depends on androgen receptor (AR) signaling to drive proliferation[2]. As a hormone-driven transcription factor, chromatin binding is a critical component of AR action, and genomic binding selectivity directly impacts AR-controlled gene expression repertoires. Recently, we reported an integrative epigenetic taxonomy of primary PCa tissues using RNA-seq coupled with ChIP-seq of histone post-translational modifications and AR, yielding a total universe of 69,330 AR binding sites (ARBS) found in 88 prostate tissues[3]. Our work revealed three distinct epigenetics-based PCa subtypes, differentiating tumors by oncogenic drivers and transcriptional programs. Other studies revealed robust programmatic plasticity in AR enhancer action during tumor development[4], metastasis formation[5], and treatment resistance[5,6], while remaining largely similar between different

metastatic lesions from the same patient[7]. Cumulatively, these reports provide evidence for PCa progression through robust, reproducible, and programmatic epigenetic reprogramming[4–7].

In this work, we investigate the extent and consequences of inter-tumor heterogeneity in AR chromatin binding using our aforementioned ARBS universe[3]. We identify a high level of AR chromatin binding heterogeneity between different primary PCa samples, with <5% of all AR binding sites (ARBS) shared by half of the tumors analyzed, which rigorous QC analyses support. Through computational integration of clinical data and a rich spectrum of PCa genomic datasets[4,5,8,9], including somatic mutation data from 200 primary[10] and 101 metastatic PCa tumors[11], functional AR enhancer activity mapping of 20,790 ARBS through massive parallel reporter assays[12], enhancer CRISPR screening[13], single-cell chromatin accessibility sequencing data from PCa cells[14] and 3D-genome data[5,14] (Supplementary Table 1), we deeply characterize the biological basis and consequences of AR enhancer heterogeneity in PCa and assess its clinical impact on patient metastatic progression.

## Results

### AR enhancer usage is highly heterogeneous in primary PCa

Previously, we reported a total universe of 69,330 ARBS in a cohort of 88 primary prostate cancers with a mean tumor cell percentage >80%, averaging 7394 peaks per tumor with FRiP scores >1.5 (Supplementary Tables 2–4)[3], which were processed in a standardized manner minimizing cell death and optimizing sample quality (see the "Methods" section). To annotate inter-tumor heterogeneity of enhancer usage, we ranked ARBS based on detected peaks in the fraction of tumors analyzed, revealing an unexpected high level of AR enhancer heterogeneity between tumors (Fig. 1a), with typical AR-inducible genes FKBP5 and KLK3 regulated by highly ranked ARBS enhancers 129 and 343 (enhID, Source Data), respectively. Based on ARBS prevalence in patients, we binned these ARBS in three categories: shared (SH; in AR sites identified in 68% or more of the patients), partially shared (PS; AR sites found in 2–67% or more of the patients) and unique peaks (UN; AR peaks observed in merely one patient).

To which degree does AR binding in frequently-used PCa cell lines represent the enhancer heterogeneity found in tumors? We over-lapped AR ChIP-seq data from treatment sensitive[8,15] and resistant[15–17] PCa cell lines with all ranked ARBS, displaying enrichment of commonly shared ARBS for all tested PCa cell lines and AR-transduced normal prostate LSHAR cells[4]. However, as a control for prostate selectivity, we observe that overlap of ARBS from PCa was largely absent in monocytic THP-1 cells[18] and molecular apocrine ER−/AR+ MDA-MDB453 breast cancer cells[19](Fig. 1b, c, Supplementary Table 5). Of note, we observe a further shift towards SH-ARBS enrichment in the acquisition of therapy resistance for both bicalutamide-resistant LNCaP[BR] and enzalutamide-resistant LNCaP derivative 42D[ENZR]. In normal epithelial prostate tissues (n = 15), we observed a highly similar heterogeneous ARBS ranking (n = 27,850, Fig. 1b, Supplementary Fig. 1A). Tumor and normal ranked ARBS rankings follow strikingly similar distributions considering the genetic heterogeneity of tumors[10,11], suggesting that AR enhancer heterogeneity is not tumor-intrinsic, but instead patient-intrinsic (Fig. 1d, Supplementary Fig. 1G), corroborating our previous case-study identifying high inter-metastatic AR binding overlap from the same patient[7].

As ChIP-seq data quality and peak calling can be impacted by technical limitations[20], we performed extensive quality control analyses (NSC, RSC, FrIP, GC%, MSPC, and single mapped reads/AR ChIP-seq read correlation tests, Supplementary Figs. 1, 2) to test whether the observed inter-tumor AR heterogeneity results from technical artifacts. Samples contained comparable high fractions of tumor cells (Supplementary Table 2) and although total ARBS identified per sample fluctuated, AR expression levels for each sample did not correlate with the total number of AR ChIP-seq peaks, nor did read depth (RD)

correlate with the total AR ChIP-seq peaks per sample, indicating biological variation between samples as opposed to technical variation (Supplementary Table 3, Supplementary Fig. 1E, F).

Strong ChIP-seq signals were found at SH-ARBS, while such signal on the individual patient level was weaker or absent at PS-ARBS. Interestingly, we identify UN-ARBS with clearly distinguishable signal from the background and of generally comparable intensity as PS-ARBS, showing that peak calling correctly identified UN-ARBS (exemplified in Fig. 1e, aggregate peaks quantified in Fig. 1f). Moreover, we detected occurrence of UN-ARBS in PCa cell lines (Fig. 1b) and alternative multiple sample peak calling (MSPC, see the "Methods" section) of ranked ARBS with multiple testing correction confirmed heterogeneous peaks as highly significant true-positives with comparable GC content, ARBS peak distribution and specific UN-ARBS signal over background (Supplementary Fig. 1B, C, E, H). Samples with low RD according to ENCODE4 Transcription Factor (TF) ChIP-seq standards had similar UN-ARBS and AR motif score distributions as those coming from samples with high RD (Supplementary Fig. 2). Moreover, no correlation was observed between samples with different RD quality categories and FRiP, NSC or RSC metrics (Supplementary Fig. 2A, C). Finally, UN-ARBS in quality outlier sample P349T did not contribute significantly to overall analyses (Supplementary Fig. 1E and Supplementary Fig. 2F, G).

As expected and previously reported[3,4,21], ~80% of ARBS are present in introns or intergenic regions that are generally considered putative enhancers with cis-regulatory potential[22] (Fig. 1g, Supplementary Fig. 1D). Interestingly, genomic distributions of ARBS were not equally distributed over consensus, with higher promoter enrichment in more heterogeneously occupied ARBS. Moreover, in all three ARBS categories, we found motifs for AR as well as canonical AR-interactors FOXA1[23] and HOXB13[4] (Fig. 1h, i, Supplementary Fig. 2H). Additionally, distributions of AR motif scores detected in UN-ARBS using MISP motif screen were equal for RD categories (Supplementary Fig. 2I). We tested our ARBS universe for significant overlaps in GIGGLE[24] (see the "Methods" section), which analyses over 14,000 individual ChIP-seq databases for TF binding overlap, and confirmed binding of these classical prostate lineage TFs for SH- and PS-ARBS (Supplementary Fig. 1H). Cumulatively, these data support genuine enrichment of heterogeneous ARBS in sequencing analysis, with co-enrichment of TFs associated with canonical AR action. Interestingly, MED1 and RNA Polymerase II subunits, but not AR nor its classical interactors, were enriched in UN-ARBS in GIGGLE (Supplementary Fig. 1H). Like PS-ARBS, UN-ARBS are mostly associated with active TSS and enhancers (Supplementary Fig. 1J), whereas this analysis was uninformative for SH-ARBS due to the relatively small group size (n = 1201). Taken together, these GIGGLE analyses indicate occupancy by functional enhancer-binding proteins on patient-unique ARBS and stress their context-dependent nature.

### Ranked ARBS have functional divergence on enhancer activity and mutation frequency

We tested ranked ARBS for bona fide enhancer activity and hormone dependency using available data from a massive parallel reporter assay testing enhancer potential for 7422 ARBS in LNCaP[25] in vehicle versus DHT conditions (Fig. 2a), resulting in ARB's enhancer potential that could be classified as inactive, inducible or constitutively active. Most-commonly shared ARBS were enriched for hormone-dependent enhancer activity (n = 286), relative to constitutively activate (n = 463) or inactive sites (n = 2467), suggesting hierarchical functional consequences of ARBS heterogeneity (Fig. 2b, c). The total set of 7422 ARBS analyzed in STARR-seq was expanded to 20,790 regions using a machine learning-based ARBS annotation, confirming our original conclusions with a more complete representation of total ARBS heterogeneity (Fig. 2b, c)[25]. To confirm these results, we performed additional STARR-seq targeted at 2495 randomly sampled heterogeneous

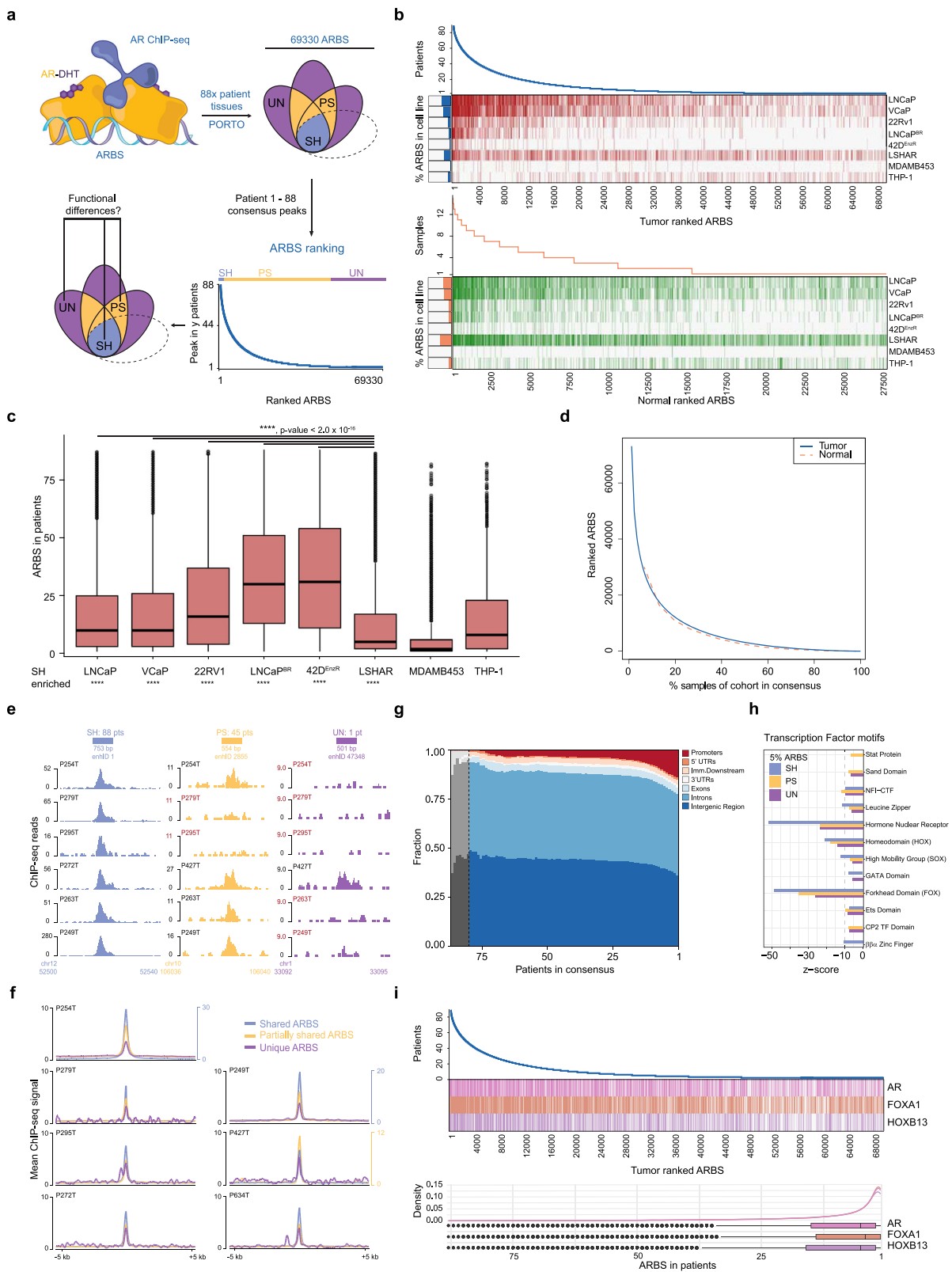

ARBS to validate predictions of heterogeneous enhancer activity and confirmed the presence of active enhancers in heterogeneous ARBS, suggesting that a subset of these ARBS function as enhancers (Fig. 2b, c, Supplementary Fig. 3A, B). Finally, individual ARBS activity was validated for a subset of regions represented in the STARR-seq library for each element designation (inducible, constitutive and inactive) through luciferase assays, which confirmed their previously identified activity

status and hormonal dependency (Supplementary Fig. 3C, D, Supplementary Table 6).

Having established an association of AR enhancer heterogeneity in PCa with biological consequences on enhancer activity, we next investigated the impact of 764 risk single nucleotide polymorphisms (rSNPs)[26–28] and single nucleotide variation (SNV) reported previously in primary PCa ($n = 278,209$)[10] and in metastatic PCa (mPCa,

**Fig. 1 | AR enhancer usage is heterogeneous in primary and normal PCa tissue.**
**a** Schematic overview: AR ChIP-seq identifies 69,330 AR Binding Sites (ARBS) in 88 patient tumor tissues ranked and binned on prevalence in patients, shared sites observed in 60–88 patients (SH-ARBS, blue), partially shared sites observed in 2–59 patients (PS-ARBS, yellow) and unique sites observed in 1 patient (UN-ARBS, purple). **b** ARBS ranking for 88 primary PCa tumors (blue, $n = 69,330$) with tumor ARBS presence in a panel of AR$^+$ cell lines (red) and ARBS ranking for 15 normal prostate epithelium (orange, $n = 27,500$) with normal ARBS presence in a panel of AR$^+$ cell lines (green). Sidebars indicate the percentage of cell line ARBS found in primary PCa tumors. **c** Boxplot quantification of primary tumor ranked ARBS ($n = 69,330$) presence in cell lines. Centerline, median; upper and lower quartiles; whiskers, $1.5 \times$ interquartile range; points, outliers. Two-tailed Student's $t$-test of means compared to LSHAR cells, ****$p < 0.0001$. Enrichment in SH-ARBS calculated by hypergeometric test, ****$p < 0.0001$, non-significant ns. **d** ARBS ranking normalized for a number of samples, comparing primary tumor (blue) and normal epithelium (orange). **e** ChIP-seq signal examples for peaks in ARBS categories, SH-ARBS enhID 1 which occurs in 88/88 patients, PS-ARBS enhID 2855 and observed in 45/88 patients and UN-ARBS enhID 47,348 in 1/88 patients. **f** Genome-wide AR ChIP-seq intensities for ARBS categories in individual tumors, line and secondary $y$-axis in blue on the right: SH-ARBS, line and secondary $y$-axis in yellow on the right: PS-ARBS, line in purple: UN-ARBS. **g** Genomic location distribution of ARBS for consensus of ARBS over all 88 tumors, gray: unstable consensus due to small amount of ARBS. **h** Transcription factor motif family enrichment at top 5% (SH), middle 5% (PS), and bottom 5% ARBS (UN) with $z$-score indicating prevalence. **i** Transcription factor motif presence for AR, FOXA1, and HOXB13 at ranked ARBS (top) and distribution across ranked ARBS (bottom). Centerline, median; upper and lower quartiles; whiskers, $1.5 \times$ interquartile range; points, outliers. Source data are provided in Source Data.

$n = 1,048,576$)[11] on primary ranked ARBS as SNVs accumulate during PCa progression. Moreover, we included germline allelic imbalance SNP data (cQTL, $n = 4454$; $q < 0.05$) which was called from our primary AR ChIP-seq data in a recent Cistrome-Wide Association Study (CWAS)[29]. Overlap of rSNPs with primary ranked ARBS was limited (52 out of 764 unique rSNPs) without any enrichment on particular ARBS, while germline cQTL SNPs and somatically acquired SNVs in primary and metastatic PCa are enriched in primary SH-ARBS ($n = 1201$, Fig. 2d, e), and these sites were previously associated with AR occupancy[30].

Highly actively transcribed regions are characterized by high levels of clustered H3K27 acetylation regions, referred to as 'super-enhancers'[31,32], and describe tissue-specific-binding profiles that are typically AR-positive in PCa cell lines and tissue. These super-enhancers encompass ARBS found throughout ranked ARBS (Fig. 2f), as exemplified by recently reported VCaP SEs at PCAT1/2 regulating MYC expression during CRPC[33], which is constituted by PS- and UN-ARBS that are affected by rSNPs and SNVs (Fig. 2g). Collectively these data show that ranked ARBS expose hierarchical enhancer activity, and display enrichment of primary and metastatic SNVs in SH-ARBS. Contrarily, super-enhancers are found scattered throughout the ARBS ranking, suggesting that PS- and UN-ARBS can functionally drive oncogenic processes.

**Enhancer-specific copy number alterations at heterogeneous ARBS drive transcriptional output in metastatic disease**
Next to SNVs, large structural events like copy number alterations (CNAs) are frequently observed in mPCa as drivers of progression[11,34]. In metastatic disease, CNAs of enhancers with tumor-driving potential for AR[11,13,35], FOXA1[36], and HOXB13[5] have been reported. A previously reported large sequencing database of 101 mPCa tissues[11] reported that 23 mPCa patients had known CNA gains at exclusively the AR enhancer locus, of which PS- and UN-ARBS were previously identified as tumor-associated[4] and metastatic-associated[5] ARBS (Fig. 3a, Supplementary Table 7, Source Data). We reconfirmed previously reported promoter–enhancer interactions by integrating ranked ARBS using H3K27ac Hi-ChIP data[5,37]. Numerous ARBS are in close proximity to a single promoter in 3D genome space, confirming previously reported VCaP AR ChIA-PET data[38]. For example, we identify chromatin loops between PS- and UN-ARBS and the AR promoter (Fig. 3a). As expected, large CNA gains and losses in mPCa patients[11] affect ARBS with enhancer–promoter interaction irrespective of ranking, including PS- and UN-ARBS, that potentially affect the expression of tumor-driving genes (Fig. 3b, Supplementary Fig. 4). Moreover, we observe a multitude of structural variations (SVs) in these sites, most notably deletions, inversions or tandem duplications at well-described PCa SV loci like tumor suppressors SPOPL at chr2q and DCC/BCL2 at chr18q (Fig. 3b).

To globally assess tumor-driving enhancer amplifications and losses, we used the cancer gene dependency repository (DepMap)[39] and investigated whether CNA-affected ARBS interact with gene loci regulating essential genes in PCa cell lines (VCaP, 22Rv1, and LNCaP, Supplementary Table 8). Based on these analyses, we found 25 essential genes interacting with CNA-affected ARBS, including previously described PCa drivers AR[11,13], GRHL2[40], and HOXB13[5] (Fig. 3c). These genes interact with PS- and UN-ARBS with a majority of ARBS found in <20 patients (Fig. 3d). Importantly, CNAs at exclusively these ARBS frequently altered corresponding mPCa gene expression[11] to a comparable extent as CNAs at exclusively gene coding sequences (Fig. 3e), as was evident for AR-upregulated (AR and ZBTB10) and AR-downregulated (IDI1, CITED2, and BCCIP) genes. These findings underline how mPCa CNA-affected ARBS are only found in a minority of primary PCa tissues, which later during mPCa have transcriptional consequences for critical PCa tumor-driving genes.

**Transcriptional variation is associated with less-commonly shared ARBS**
We predicted which ARBS influence gene expression most in our cohort by modeling H3K27ac-based HiChIP interacting ARBS–promoter pairs and matched gene expression from 88 primary patients in a generalized linear model (GLM), which generalizes linear regression of ARBS occupancy in patients to their transcriptional response. For example, the complete ARBS landscape of CITED2 (PCa tumor-driving gene, Fig. 3e) has a high degree of ARBS heterogeneity including many PS- and UN-ARBS scattered between individual tumors on Chr6 (Fig. 4a).

The CITED2 GLM shows that an ARBS found in 54 primary tumors, and not the most-commonly shared site found in 80 tumors, was most significantly associated with gene expression differences (Fig. 4b), with model assumptions such as linearity of points and limited individual point influence holding (Supplementary Fig. 5A). Additionally, as expected from trends observed in genome-wide AR ChIP-seq peak strengths in the three ARBS categories (Supplementary Fig. 1E, F), we find a clear negative correlation between ChIP-seq peak strength and ARBS category across all ARBS and patients (Supplementary Fig. 5B, C). In total, 2026 ARBS regulating 1901 unique genes were identified by GLMs to significantly associate genome-wide AR ChIP-seq with expression differences ($p < 0.001$).

As these observations are from bulk measurements, single-cell assay for transposase-accessible chromatin sequencing (scATAC-seq) from LNCaP[14] was used to infer promoter–enhancer interactions and ARBS accessibility throughout cell cycle phases. Although few ARBS were cell-cycle phase-specific in arrested bulk LNCaP[41] and AR activity fluctuates among scATAC-seq clusters, there is an enrichment of ranked ARBS and de novo AR and FOXA1 motifs in differentially accessible chromatin (Supplementary Fig. 6). Links are observed between most heterogeneously bound ARBS and the CITED2 promoter in LNCaP (Fig. 4c), in agreement with LNCaP H3K27ac Hi-ChIP data[5].

Subsequently, for CITED2 we assessed which proximal ARBS have a functional impact on transcriptional output. For this, we genetically

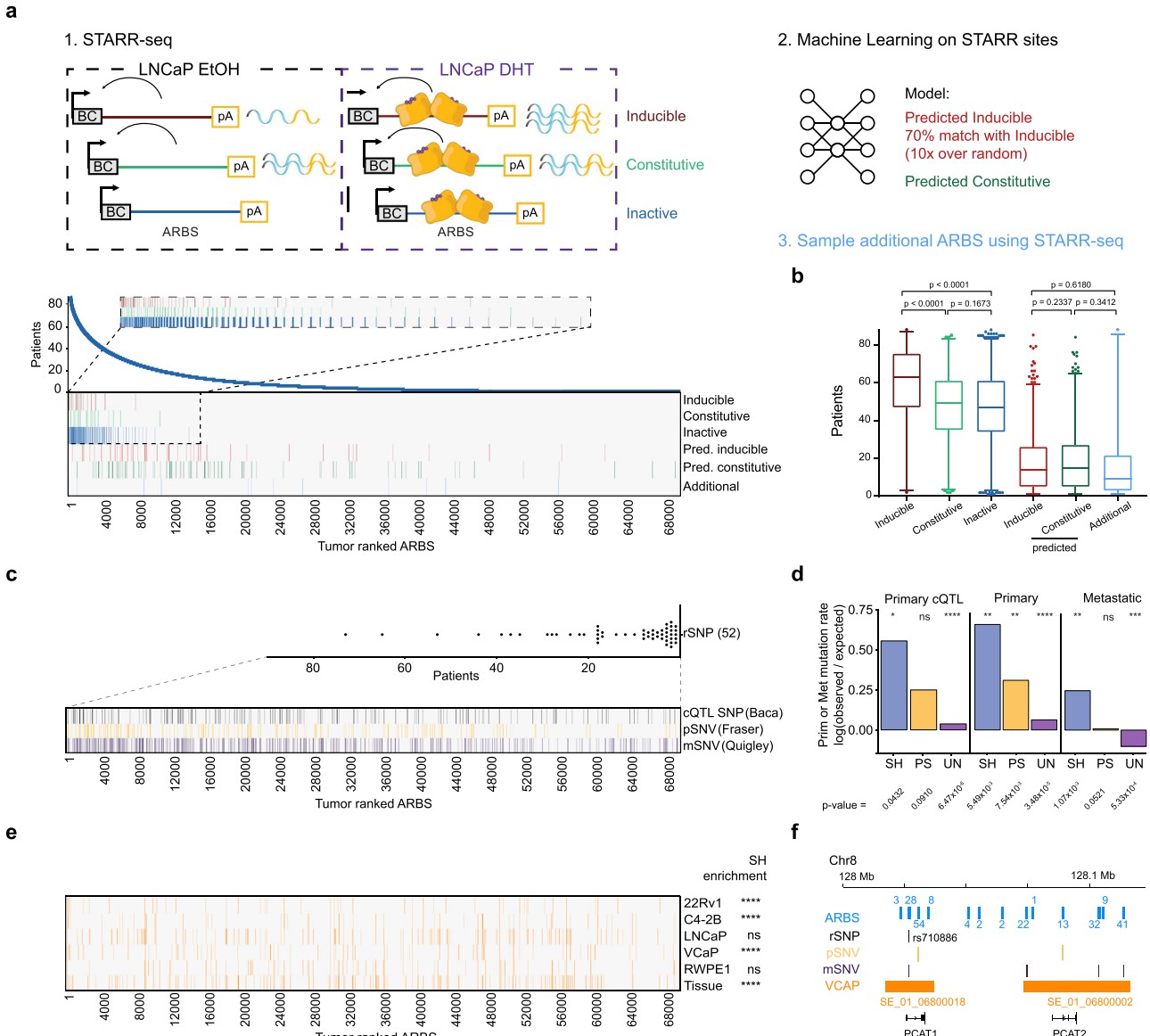

**Fig. 2 | Ranked ARBS have functional divergence on AR enhancer activity, mutational frequency, and presence of super enhancers in cell lines and tissue.**
**a** Enhancer activity for ranked ARBS found in EtOH over DHT LNCaP conditions in STARR-seq, machine learning predicted ARBS, and the second set of designed ARBS. Inset: zoomed-in STARR-seq regions for ARBS ranked 1–15,000.
**b** Distribution of ARBS with enhancer activity identified in STARR-seq experiments. Inducible $n = 286$, constitutive $n = 463$, inactive $n = 2467$, predicted inducible $n = 1237$, predicted constitutive $n = 1671$, active $n = 149$ ARBS. Two-tailed Student's t-test of means. Centerline, median; upper and lower quartiles; whiskers, 1.5× interquartile range; points, outliers. **c** ARBS ranking with presence of PCa risk single nucleotide polymorphisms (rSNP, and single nucleotide variations (SNV) identified in primary prostate cancer (cistrome Quantitative Trait Locus, cQTL; primary SNV,

pSNV) and metastatic SNV, mSNV. Inset: distribution of rSNPs at ARBS among patients. Centerline, median; upper and lower quartiles; whiskers, 1.5× interquartile range; points, outliers. **d** Observed over expected background primary or metastatic mutation rate for ARBS rankings. Two-tailed Fisher's exact test on untransformed values, *$p < 0.05$, **$p < 0.01$, ****$p < 0.0001$, ns = non-significant. **e** Ranked ARBS identified at super-enhancer genomic locations for PCa cell lines and tissue as reported by SEdb, SH-ARBS hypergeometric test of enrichment, ****$p < 0.0001$. **f** Genomic snapshot of PCAT1 and PCAT2 locus on Chr8 with ARBS prevalence in patients (blue numbers), rSNP rs710886 (black), primary SNVs (yellow), metastatic SNVs (purple), and VCaP SE (names in orange) presence. Source data are provided in Source Data.

deleted the entire ARBS through CRISPR-Cas9-mediated genome editing by transducing LNCaP with Cas9 and confirming its activity in a polyclonal population (Supplementary Fig. 7A, B), designing guide pairs and combining these in pools to maximize the chance of a KO in case of a non-effective single guide. Guide pairs and pools were targeted to ARBS edges identified with scATAC-seq, which had AR binding in LNCaP (Fig. 4d) and we confirmed successful Cas9 deletion through genomic DNA PCR. We observed a concomitant drop in CITED2 expression for the CITED2 enhancer found in 54 primary tumors (CI54), which was predicted to most significantly affect

transcription, although effect sizes varied between guide pair and guide pools (Fig. 4e, Supplementary Fig. 7D). Additionally, as an orthogonal method for Cas9 deletion, we employed CRISPR interference through a modified Suntag-system[42] which enables recruitment of 10 repressive KRAB effectors at a locus through sgRNAs (Suntag-KRAB, validated in Supplementary Fig. 7A, C). Suntag-KRAB repression at the CITED2 enhancers confirms that CI54 most significantly affects transcription (Supplementary Fig. 7F), while Cas9-mediated CI54 deletion in the CITED2 expressing AR⁺ breast cancer cell line MDA-MB453 did not result in a transcriptional decrease, as this cell

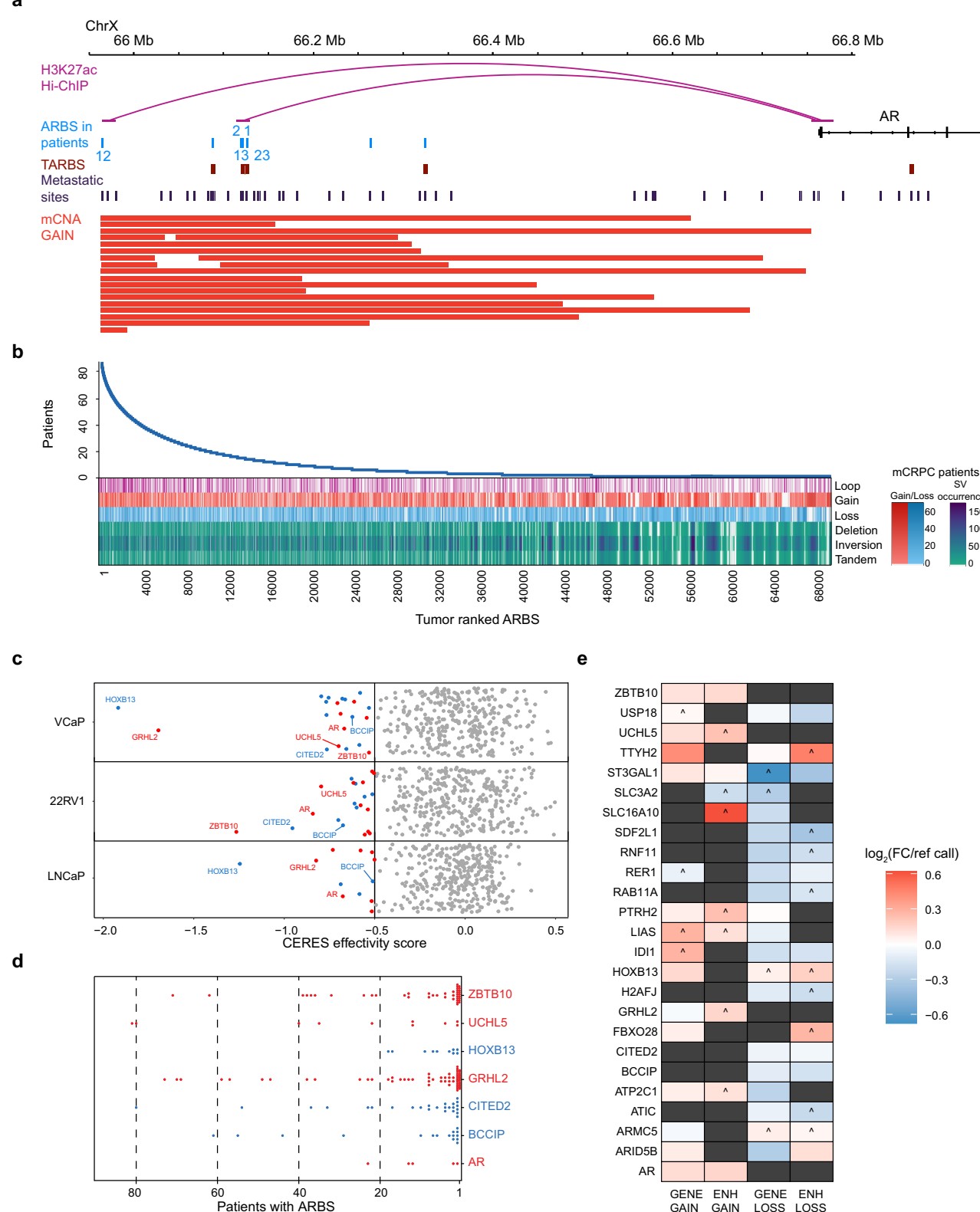

line does not use enhancer CI54 (Supplementary Fig. 7A, G, H). These data suggest that less-frequently shared ARBS can have the largest impact on transcriptional output, which could be attributed to varying degrees of functional redundancy among ARBS.

To further confirm these findings, we focused on the AR locus, for which numerous inter-tumor heterogeneous ARBS loop to the TSS

(Fig. 3a) and with confirmed AR binding and transcriptional activity in our primary patient cohort (Supplementary Fig. 8A). Using two orthogonal CRISPR drop-out screens tiling 878 sgRNAs across the entire AR enhancer region on ChrX[13], AR13 (found in 13 primary tumors) proved most-critical for tumor cell proliferation, in contrast to the more-commonly shared AR23 or less-common AR2 (Fig. 4f, g).

**Fig. 3 | Enhancer-specific copy number alterations at ranked ARBS affect transcription at interacting PCa-dependent genes in metastatic disease.** **a** Genomic snapshot of enhancer region upstream of AR locus with H3K27ac Hi-ChIP interaction data, number of primary patients with ARBS, primary tumor ARBS (TARBS), metastasis-associated ARBS and metastatic patient enhancer-specific copy number gains (23/101 patients). **b** ARBS ranking showing H3K27ac Hi-ChIP interactions with gene promoters and ARBS affected by copy number gains and losses and SVs such as deletions, inversions, and tandem duplications. False color scale for CNAs and SVs indicate occurrence. **c** PCa cell line VCaP, 22Rv1, and LNCaP gene dependencies (CERES effectivity score) for interacting ARBS affected by copy number gains (red) and loss (blue). The predominant CNA is shown, i.e. copy number gains for oncogenes occur at much higher frequencies than losses. **d** ARBS interacting with promoter of essential PCa genes, plotted for prevalence in patients with color denoting predominant copy number gains (red) and losses (blue) at these loci. **e** Metastatic PCa patient RNA-seq $\log_2$fold expression changes over copy number neutral samples (ref call) for patients with CNAs exclusively at ARBS or exclusively at gene coding sequences. ^ denotes only single patient expression value in mPCa cohort, dark gray not present in mPCa cohort. Source data are provided in Source Data.

Jointly, these data further confirm that heterogeneous ARBS can impact cellular fitness to a larger degree than more commonly shared ARBS. Interestingly, exclusively AR13 shows strong HOXB13 motif enrichment (Supplementary Fig. 8B), and HOXB13 was detected at AR13 for LNCaP and 22Rv1 cells using ChIP-seq[15] (Supplementary Fig. 8C), providing a possible explanation for these observations given HOXB13's critical nature in regulating AR-transcriptional function[5].

To further investigate this, we confirmed the differential proliferation effects of AR23 and AR13 enhancer perturbation using LNCaP:Suntag-KRAB and observed similar trends in proliferation defects (Supplementary Fig. 8D, Fig. 4f, g), with AR13 perturbation having the biggest impact on cell proliferation as opposed to AR23, underlining the impact of heterogeneous ARBS on cellular fitness. Finally, we performed HOXB13 ChIP-qPCR on LNCaP:Suntag-KRAB with either NT or AR13 sgRNAs and observed that perturbation of AR13 through KRAB-mediated heterochromatinization leads to a sharp decrease of HOXB13 binding at this locus compared to NT (Supplementary Fig. 8E). These results confirm that HOXB13 binding at AR13 influences PCa cellular fitness through enhancing AR transcription.

## Metastasis-associated heterogeneous ARBS in poor-outcome primary tumors drive tumor-promoting gene expression pathways

Prior studies have identified ARBS found selectively enriched in normal tissue (NARBS) over primary tumors (TARBS)[4], metastasis-associated sites (met-ARBS), or those found in primary PCa[5] and ARBS linked to good and poor outcomes[8]. As such, ARBS alterations specific to different states of PCa progression and disease outcomes have been established. With TARBS representing a general feature of primary PCa, we observe an expected TARBS enrichment for SH-ARBS, with NARBS poorly represented in our tumor samples. In contrast, met-ARBS are found mostly in heterogeneous ARBS, suggesting that heterogeneous ARBS contribute to disease progression (Fig. 5a, Supplementary Fig. 9A). In agreement with this observation, good outcome ARBS are more prevalent at SH-ARBS as compared to poor outcome ARBS (Supplementary Table 9). Our 88-patient cohort was designed as a case-control study based on biochemical recurrence (BCR)[3], enabling us to independently confirm the clinical implications of AR enhancer heterogeneity. We observed a significant difference between cases and controls in good/poor outcome ARBS ratios (Fig. 5b, Supplementary Fig. 9B ratios good:poor: (1) >1.2 good, (2) 1.2 > mixed > 0.8, (3) poor < 0.8), independently confirming poor outcome ARBS being more-heterogeneously distributed among tumors and highlighting the predictive power of these previously reported sites.

Finally, we investigated whether heterogeneous ARBS plays a role in PCa progression to metastatic disease. We observed a striking enrichment of met-ARBS at PS- and UN-ARBS over primary disease TARBS (Fig. 5c). Moreover, we separated TARBS and met-ARBS based on their presence in patients with a high chance of BCR (case) or with a low chance of BCR (control), and observed a significant enrichment of met-ARBS in cases over control patients in PS- and UN-ARBS, whereas no difference in TARBS enrichment is found in both patient populations (Fig. 5d).

We observed that met-ARBS were selectively enriched in PS- and UN-ARBS in patients whose tumors ultimately progressed. To confirm these observations on the transcriptional level, we calculated Gene Ontology Regulatory Potential (GO-RP) scores of the bottom 10% of case-specific and control-specific met-ARBS (RPscore > 0.05, Supplementary Fig. 9C, D), identifying distinct sets of genes (Fig. 5e) representing different kinds of pathways (Fig. 5f). GO-RP uses not the only distance between enhancers and promoters, but also adjusts these scores and ranks elements based on the integration of ChIP-seq and expression data to accurately identify target genes. Notably, heterogeneous case met-ARBS regulate hallmarks of cancer pathways involved in cholesterol synthesis[43], mTORC1 signaling[44], androgen response, and WNT beta-catenin signaling[44], whereas the P53 pathway, which is often inactivated in mPCa[10,45], was activated by heterogeneous control met-ARBS. Moreover, individual metastasis-promoting genes such as proto-oncogene RET[46] or migration and invasiveness-related genes like CDH17, CDH18, ITGB5, and ITGB7, or osteoclast-promoting TF FOS2L[47] involved in the formation of bone metastases are found in this set and are regulated by heterogeneous met-ARBS detected in cases. With comparable enrichment for TF motifs for both groups (Fig. 5g), transcriptomics data from Taylor[48] and Grasso[49] cohorts show that the most-heterogeneous ARBS found in primary tumors that ultimately relapse regulate genetic programs selectively altered in mPCa (Fig. 5e, h, Supplementary Fig. 9E).

## Discussion

Ranging in the order of 20,000–70,000 ARBS for PCa cell lines[15–17] or tumors[3–5], the collection of experimentally reported ARBS is smaller compared to the total number of AR consensus motifs found throughout the human genome[50]. Indeed, AR chromatin binding requires a permissive epigenetic environment for AR-modulated gene expression, and reproducible epigenetic changes in PCa disease state transitions are associated with AR binding plasticity at these sites[4,5]. While highly recurrent state-specific alterations in AR chromatin profiles are gradually becoming established, inter-tumor heterogeneity of AR enhancer action remains largely unexplored.

We report a high level of ARBS heterogeneity between PCa primary tumors, which occurs in the same proportion in normal prostate epithelium. Such ARBS heterogeneity has biological consequences and clinical implications. First, while commonly shared ARBS are enriched for somatic mutations and exhibit AR-driven functionality, ARBS that are associated with biochemical recurrence and are thus clinically relevant were specifically acquired in metastatic disease (met-ARBS), uncommon in the patient population, and often unique to patients. As only ~30% of primary PCa patients show tumor relapse after radical prostatectomy, the majority of patients do not progress postoperatively[51]. In agreement with this, ARBS enriched in metastatic disease were not observed in the commonly shared fraction of primary ARBS. Second, since UN-ARBS are frequently depleted for enhancer activity in hormone-responsive LNCaP STARR-seq, heterogeneous metastasis-associated ARBS in primary PCa may lack transcription complex components essential for selective enhancer function before activation during progression, in line with our previous observations on AR plasticity in mCRPC[5]. Third, although SNVs at ARBS rarely alter

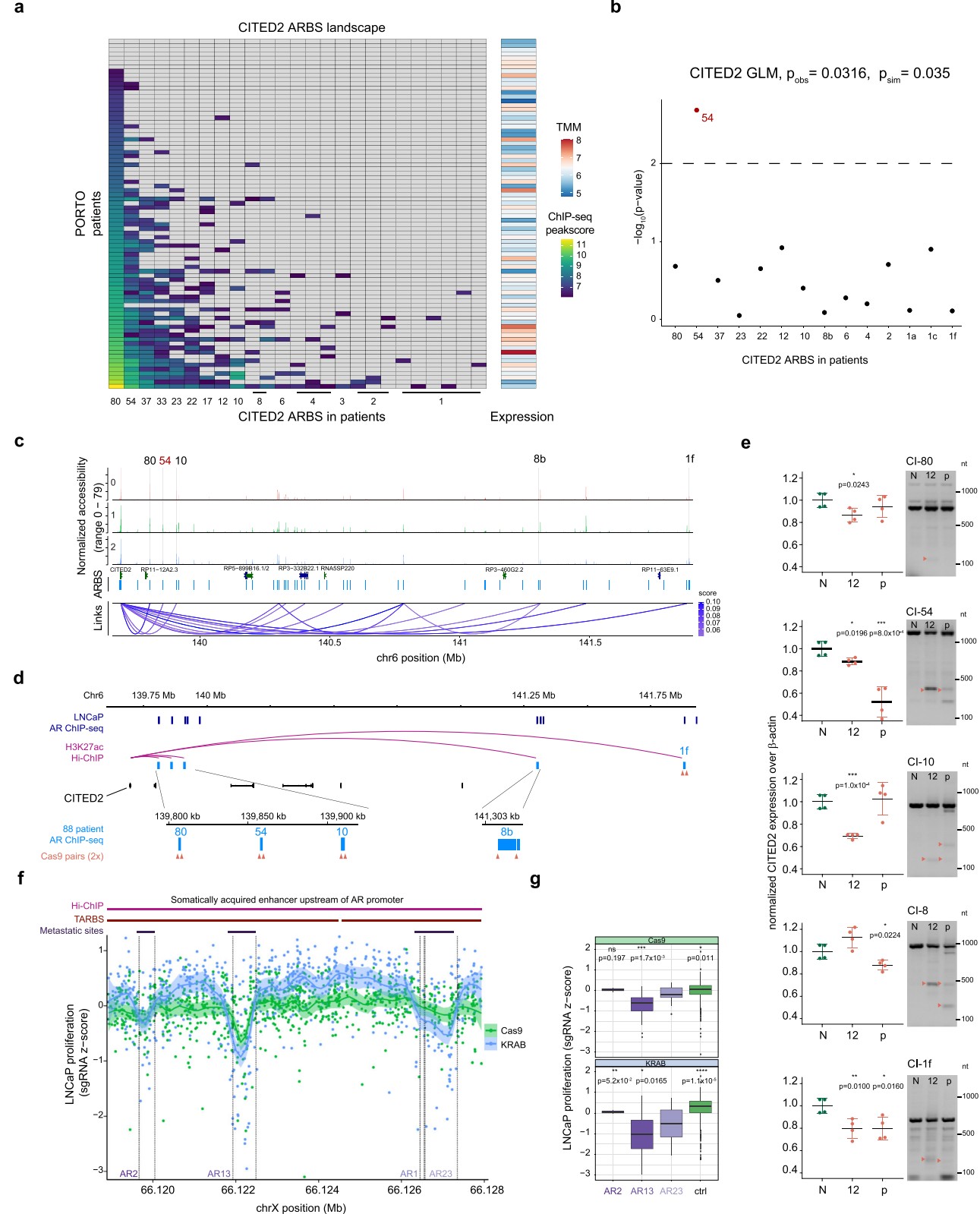

the linked gene's expression[33], here we show that large-scale CNAs in mPCa affect a subset of ARBS linked to the expression of genes on which PCa cell lines are dependent. Such ARBS may not be apparent in primary PCa, but later during disease progression contribute strongly to mPCa phenotypes as illustrated by the regulation of clinically relevant genes such as RET[46] or FOSL2[47] that separate primary from metastatic disease in independent patient cohorts. Finally, AR

enhancer plasticity is affected by protein-coding somatic mutations, as described for FOXA1[52] and TMPRSS2-ERG fusions[53]. Importantly, the resulting distinct AR chromatin interaction profiles have been linked with resistant PCa phenotypes, which switch ARBS usage under the pressure of potent AR inhibition[14,54].

Other potential underlying biological causes for ARBS heterogeneity include pioneer factor FOXA1 mutations that alter its

**Fig. 4 | Transcriptional variability is associated with less-commonly shared ARBS. a** Primary PCa tumor AR ChIP-seq log-transformed MACS peak scores for ARBS with H3K27ac HiChIP interaction with CITED2 gene promoter. Gray: no ARBS detected in AR ChIP-seq, Right: Matched log-transformed TMM-normalized RNA-seq expression levels for patients. **b** Generalized linear model (GLM) -log($p$-values) from fitting log-transformed MACS peak scores (predictor) with CITED2 gene expression (response), ARBS were filtered for overlapping H3K27ac presence in LNCaP. Top: Observed model $p$-value and simulated $p$-value obtained from permutation tests ($n = 1000$) based on likelihood ratio test. **c** LNCaP single-cell chromatin accessibility for three cell clusters at CITED2 locus with filtered CICERO co-accessibility scores of links for CITED2 enhancers in 80, 54, 10, 8, and 1 patient(s). **d** Genomic snapshot of CITED2 locus with LNCaP AR ChIP-seq, ranked ARBS from tissue ChIP-seq with CITED2 H3K27ac Hi-ChIP promoter–enhancer pairs found in 80, 54, 10, 8, and 1 patient(s) and design of Cas9 sgRNA pairs (orange, 2 sgRNAs per arrow). **e** Normalized expression levels of CITED2 over β-actin as measured by RT-qPCR 40 days after infection with non-targeting control (N), sgRNA pair 12 (12), the pool of all sgRNAs guides (p) with gDNA PCR verification of cas9 cut from the same isolate for CITED2 interacting ARBS found in 80, 54, 10, 8, and 1 patient(s). Orange arrows denote cut DNA fragments, nt = nucleotide weight. Representative experiment, center line, mean; error bars, SD; two-tailed Student's $t$-test of means on technical replicates, *$p < 0.05$, **$p < 0.01$, ***$p < 0.001$. **f** LNCaP proliferation $z$-score for 878 sgRNAs targeted at AR enhancer locus in Cas9 perturbation or dCas9-KRAB inhibition tiling assay with dotted lines denoting ranked ARBS in 2, 13, 1, and 23 primary tumors. Shaded areas denote 95% confidence interval. **g** LNCaP proliferation $z$-score for sgRNAs in ARBS found in 2, 13, 1, and 23 primary tumors, with control comprising $z$-scores of all other sgRNAs in this region. sgRNAs per ARBS, AR2: 29, AR13: 49, AR23: 49, AR1: 4, ctrl: 1573. Centerline, median; upper and lower quartiles; whiskers, 1.5× interquartile range; points, outliers. Two-tailed Student's $t$-test of means with AR23 as reference group, *$p < 0.05$, **$p < 0.01$,***$p < 0.001$ ****$p < 0.0001$, ns non-significant. Source data are provided in Source Data.

cistrome[55–57], disease stage-specific epigenetic states (either induced by therapy or not)[4,5,58], and clonal heterogeneity arising from tumor multifocality[59,60]. As such, the heterogeneity of ARBS present in primary PCa could be seen as a potential pool of patient-intrinsic ARBS driving clinically relevant phenotypes by influencing transcription in later-stage disease, emphasizing the clinical impact of ARBS variations among tumors. In contrast, we observed a shift towards SH-ARBS in bicalutamide and enzalutamide-resistant PCa cell lines, underlining how currently available cell line models do not recapitulate the clinically observed ARBS heterogeneity.

Interestingly, AR enhancer chromatin accessibility varies between specific cell-cycle phases[41], and in our analysis of scATAC-seq data, which suggests that cell-cycle differences in accessibility could form another partial source of ARBS heterogeneity. Additionally, we limit our focus on ARBS that most significantly impact the expression of their target genes, while recent work shows enhancer cooperativity and mechanisms of synergism in modulating transcriptional output[61], warranting further study to dissect individual contributions of heterogeneous ARBS in enhancer-promoter landscapes. Indeed, genes are regulated by different numbers of enhancers acting at varying levels of redundancy and genomic distances, further complicating the cross-comparison of enhancer influence between individual genes.

Along similar lines, we grouped ARBS into three categories as we reasoned that SH-ARBS should occur in more than two-thirds of our cohort while UN-ARBS are defined by occurrence in a single patient. However, upon shifting the definition of SH-ARBS to occurrence in half of our cohort ($n = 3113$), our conclusions remain similarly statistically supported. Although we excluded false negatives and positives to the best of our abilities, some UN-ARBS could in fact be PS-ARBS or vice versa, as peaks could have been erroneously called. Importantly, such occurrences would not significantly alter our analysis of clinical consequences as this analysis is based on the lowest decile of ARBS rather than only UN-ARBS, and as such underline the robustness of conclusions drawn from categorizing ARBS. Finally, we employ machine learning to classify patient ARBS in a grouped analysis using LNCaP STARR-seq[12], which has limitations to interpretability due to contextual differences between patients, constructs, and cell lines. However, we have recently observed a strong correlation between LNCaP STARR-seq data and H3K27ac signals in patient samples[62], making such a predictive model relevant for clinical activity.

In summary, we report an immense level of AR enhancer heterogeneity in normal tissue and primary PCa with functional consequences on gene expression during PCa progression. We provide evidence that heterogeneous ARBS are affected by somatic mutations and that CNAs acquired during PCa progression functionally contribute to malignant phenotypes to drive different gene expression programs involved in mPCa. Critically, heterogeneous AR enhancer usage in subsets of patients distinguishes patients on the outcome, indicating that epigenetic heterogeneity may be relevant in disease progression and could provide important opportunities to the field in advancing and personalizing PCa screening or treatment.

## Methods

### Ethics statement
No new patient samples were used in this study. The AR ChIP-seq data on 88 primary prostate cancers have been described previously[3], for which informed consent and IRB approval were given.

### Supplementary table of databases used in this study
All GEO accession numbers and references of publicly available data used in this study are summarized in Supplementary Table 1.

### Sample collection
All prostate tumor samples described in this study, have previously been reported[3] and have been collected at the Department of Pathology of Portuguese Oncology Institute of Porto. Sample collection was performed in a highly standardized and optimized setup. After surgery, all radical prostatectomy samples were directly processed by the same dedicated uropathologist in a standardized manner. In brief, the whole prostate was cut transversally to the main axis into slices of ~6–7 mm thickness. Each slice was subsequently cut into quadrants and the fragment of each quadrant was halved, producing twin fragments, of which one was immediately snap-frozen and the other immediately placed in a cassette and immersed in neutral buffered formalin after which it was processed for paraffin embedding, routine histopathological assessment, and immunohistochemistry. This procedure minimized cell death and autolysis due to technical work-up, which is validated on H&E staining, as no artifacts of poor fixation were disclosed. Fragments were stained for standard neuroendocrine markers Chromogranin A and Synaptophysin and assessed for normal epithelium low expression levels (-1%). Samples were assessed for tumor cell percentage and lower tumor cell-containing samples were enriched through macro-dissection prior to cryo-sectioning to avoid significant tissue heterogeneity.

### Statistics and reproducibility
This study is designed as 97 patients with either a high or low risk of biochemical recurrence, and whose tissues had 88 AR ChIP-seq samples passing quality control[3]. SH-ARBS sample size was chosen as ARBS occurring in 60/88 samples, PS-ARBS sample size was chosen as all other ARBS except for ARBS occurring once in this cohort, which were defined as UN-ARBS.

### ChIP-seq ARBS calling, ranking, and data analysis
Chromatin Immunoprecipitation sequencing data from primary tumor tissues were called using DFilter (v1.5, bs = 50, ks = 30, refine, nonzero)[63] and MACS (v1.4, $p$-value cutoff 10e−7)[64], and only peaks identified using both algorithms were considered for further

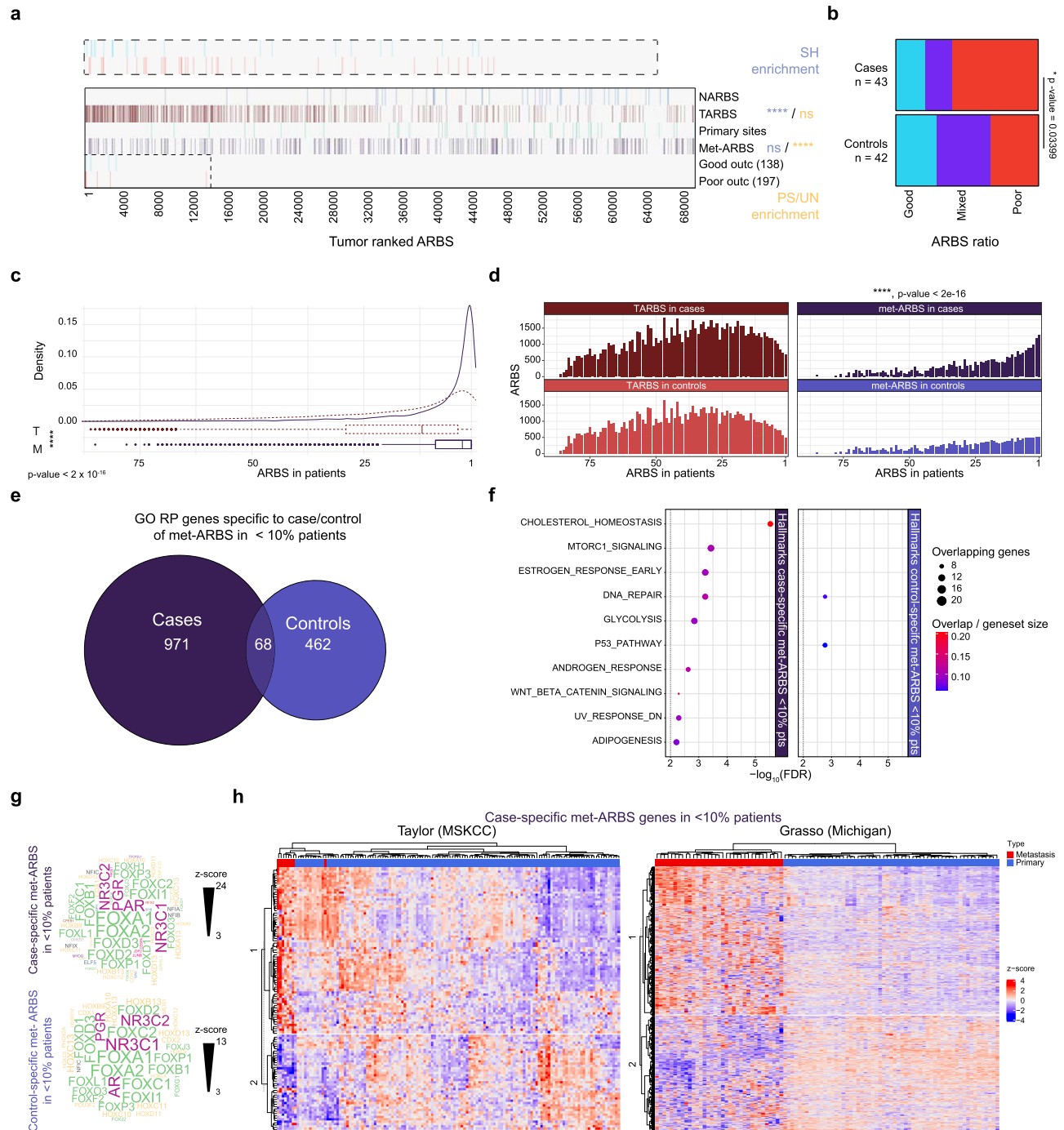

**Fig. 5 | Metastasis-associated heterogeneous ARBS in poor-outcome primary tumors drive different gene expression pathways. a** Normal tissue and tumor enriched ARBS (NARBS and TARBS, respectively), primary sites, metastatic associated ARBS (met-ARBS), good and poor outcome sites presence in ranked ARBS. SH-ARBS (blue) or PS + UN-ARBS (yellow) enrichment through a two-sided hypergeometric test of enrichment, ****p < 0.0001, non-significant ns. Inset: zoom-in on outcome sites for ranked ARBS 1 through 15,000. **b** Distribution of ratio of outcome sites per primary PCa patient split for BCR development (case) or not (control). Ratios good:poor; (1) >1.2 good (blue), (2) 1.2 > mixed > 0.8 (purple), (3) poor < 0.8 (red). Two-sided Fisher's exact test, *p < 0.05. **c** Distribution of TARBS (T, red dotted line) and Met-ARBS (M, purple line) in ranked ARBS. Centerline, median; upper and lower quartiles; whiskers, 1.5× interquartile range; points, outliers. Two-tailed

Student's *t*-test of means, ****p < 0.0001 **d** Histograms of TARBS (red) or Met-ARBS (purple) counted in patients, split for BCR development (case) or not (control). Two-sided Kolmogorov–Smirnov test of distribution, ****p < 0.0001. **e** Euler diagram of a number of genes with GO regulatory potential score >0.5 for met-ARBS which are specific for primary patients with biochemical recurrence (BCR) development (case) or without (control). **f** Gene set enrichment analysis (GSEA) for hallmarks of cancer collection using GO RP score >0.05, for genes linked to case- (left) and control-specific (right) met-ARBS in less than 10% of patients. **g** Transcription factor motif analysis of case/control specific met-ARBS with motif *z*-scores. **h** Heatmap clustering (*k* = 2) for Taylor (FDR < 0.01) and Grasso cohort *z*-score expression levels from patient tissues filtered for case-specific met-ARBS genes <10% of patients. Source data are provided in Source Data.

downstream analyses after extensive quality control[3]. All reported genomic locations in this study are in GrCh37/hg19 and when required for analysis were lifted over from GRCh38/hg38 using ucsc-liftover (v366) command-line and hg38ToHg19.over.chain file. Transcriptional Start Sites (TSS) including promoter regions were removed by excluding 1000 bp from all gene TSS in all patient ARBS peaklists using BEDtools(v2.29)[65]. ARBS peaklists were then intersected and peak occurrence in the total population was counted using BEDtools. Tumor ARBS were ranked for patient occurrence, then for genomic location, and subsequently used to intersect with other databases (Source Data) using BEDtools and plotted for the co-occurrence of ARBS in both sets in a heatmap by R package NMF(v0.23.0)[66]. ARBS were grouped into three categories based on ARBS prevalence in the entire cohort, with SH-ARBS defined as occurring in 60 or more patients (68% of patients), PS-ARBS defined as occurring in 2–59 patients (2–67% of patients) and UN-ARBS defined as occurring in only one patient. Normal prostate epithelium ARBS peaklists were pooled from two databases[4,5] and processed similarly and the tumor and normal ARBS rankings were normalized by patients for comparison (Supplementary Table 6). Differences in the spread between normal and tumor ARBS ranking in cell lines were statistically compared using two-tailed Student's $t$-test of means. Enrichment of ARBS in ranked ARBS groups was calculated using hypergeometric test with $P(X > x)$ condition.

Tumor-ranked ARBS consensus were generated sequentially with minimal ARBS overlap starting at peaks found in 88 patients and then found in one less patient for each consensus using DiffBind(v3.4)[67] and resulting genomic regions were annotated using ChIPpeakAnno(v3.28.0)[68]. Genomic peak snapshots were taken using IGV-Web(v1.6.3)[69]. ARBS ChIP-seq tornado plots and aggregate genomic ChIP-seq signal plots were generated using Easeq(v1.03)[70], bams around ARBS genomic locations were gated for ARBS categories and normalized across the genome. Multiple testing corrections and quality control of ranked ARBS as true-positives were performed using MSPC(v5.4.0)[71] with default settings. TF binding overlap of ranked ARBS per category with publicly available ChIP-seq data from ENCODE, Roadmap Epigenomics, and GTEx was performed using transcription factor GIGGLE (http://dbtoolkit.cistrome.org/)[24]. Ranked ARBS were analyzed for TF motifs using Cistrome SeqPos[72] and screened for AR, FOXA1, or HOXB13 motif presence using Cistrome MISP[73], and the validity of results was checked through independently running MEME-MAST[74] on ARBS with standard settings. Permutation tests of equality on density distributions were performed using R package sm(v2.2)[75] through function sm.density.compare() with model assumption = 'equal', $n = 1000$ permutations with ngrid = 100 for plotting estimates.

## Cell culture

LNCaP cells (ATCC, CRL-1740) were grown in RPMI medium (Gibco) with 10% fetal bovine serum and 1% penicillin–streptomycin (PS). HEK cells (ATCC, CRL-3216) were grown in DMEM medium (Gibco) with 10% fetal bovine serum and 1% penicillin–streptomycin. LNCaP and HEK cells were dissociated with 0.05% trypsin solution. For experiments in different hormonal conditions, cells were cultured for 3 days in RPMI medium containing 5% dextran-coated charcoal (DCC)-stripped FBS and 1% PS. The medium was replaced with RPMI medium containing 5% charcoal-stripped FBS, 1% PS, and 10 nM synthetic androgen R1881 or an equal amount of DMSO. After 4 h in the R1881-containing medium, cells were harvested for simultaneous total RNA and gDNA isolation. All cell lines have been authenticated through STR profiling and were regularly found to test negative for mycoplasma contamination.

## STARR-seq and luciferase validation

STARR-seq was performed and analyzed for EtOH or DHT-stimulated LNCaP cells[12], which were electroporated using a library focused on common clinical ARBS (NARBS and TARBS)[4]. Machine learning of ARBS activity was performed[12], excluding predicted inactive sites while intersecting predicted constitutively active and predicted inducible ARBS with ranked ARBS using BEDtools. Additional STARR-seq sites were designed in an unbiased manner among the entire ranked ARBS universe by randomly sampling ARBS, resulting in a library of STARR-seq constructs containing more homogeneously spread ARBS throughout the ranking than our previous targeted NARB and TARB library. Additional STARR-seq ARBS were electroporated in LNCaP and treated with either EtOH or 10 nM DHT for 4 h and harvested 72 h post electroporation. To determine additional ARBS with STARR activity we first downsampled 2 replicate LNCaP vehicle (ethanol) STARR-seq files to equivalent read counts and merged them into one file with samtools (v1.8)[76]. Kmeans clustering was carried out using deepTools (v2.0) bamCoverage, computeMatrix, and plotHeatmap functions[77]. Those ARBS with signals in the top 1 and 2 of 3 clusters were considered active ($n = 149$). All other ARBS found in cluster 3 were considered inactive ($n = 2346$).

For performing luciferase assays, regions of interest were PCR amplified from pooled male human genomic DNA (Promega) with overhangs added for Gibson assembly. The amplified regions were then cloned into a modified STARR luciferase validation vector ORI empty plasmid (Addgene #99298) using the NEBuilder HiFi DNA Assembly master mix (NEB). Primers used to amplify the regions are described in Supplementary Data 1. All insert sequences were verified by Sanger sequencing using RV Primer 4 and SV40pA-R. Then, $1.5 \times 10^5$ LNCaP cells were seeded in phenol red-free RPMI-1640 media (Gibco), supplemented with 10% charcoal dextran-stripped FBS (Gibco, US origin) without any antibiotics in PEI-coated 24 MW plates. 24 h following seeding, cells were transfected using 500 ng of reporter DNA per well in 50 µl of Opti-MEM (Gibco) along with 5 ng of pRL-CMV Renilla reporter plasmid as a transfection control, using Mirus TransIT-2020 transfection reagent (Mirus Bio) at a 1:3 DNA:transfection reagent ratio according to the manufacturer's protocols. 48 h post-transfection, cells were treated with 10 nM of DHT or an equivalent amount of 100% ethanol (vehicle control) for another 24 h prior to harvest using 200 µl of 1× passive lysis buffer (Promega). 20 µl lysate was used for each assay using the Dual Glo Luciferase Assay kit (Promega) according to the manufacturer's instructions using the M200Pro TECAN Luminometer in technical triplicate with a minimum of three biological replicates. For data analysis, technical triplicates were averaged after which firefly luciferase ($F_{Luc}$) values were normalized to the Renilla luciferase ($R_{Luc}$) values.

## Somatic mutations, mutation rate, and super-enhancers

Publicly available rSNP (Supplementary Table 1[78–80]), primary and metastatic prostate cancer SNV data (Supplementary Table 1, Source Data) was downloaded and intersected with ranked ARBS using BEDtools. Observed over background mutation rate in the whole genome was calculated and statistically compared to mutation rates in ARBS using Fisher's exact test. Super-enhancer genomic location data was downloaded from SEdb[9] and dbSUPER[81] for prostate cancer cell lines and tissues and intersected with ranked ARBS using BEDtools. Additional cQTL rSNP data was inferred from allelic imbalance observed in AR ChIP-seq sample, of which $p$-values were adjusted using R package $q$-value (v.2.28.0), with significant imbalance called when peaks contained one or more SNPs at $q < 0.05$.

## Genomic interaction data handling, CNA data partitioning, and gene dependency data

H3K27ac HiChIP sequencing regions from LNCaP cells[5] were downloaded and filtered for promoter-enhancer (PE) or enhancer–promoter (EP) interactions. Filtered anchors were intersected with ranked tumor ARBS using BEDtools. Copy Number Alteration and normalized RNA-seq data from 101 metastatic PCa patients[11] were downloaded, with

CNAs merged and intersected with ranked ARBS using BEDtools for each patient. Genomic ranges spanning copy number alterations exclusively inside gene coding sequences or exclusively at intergenic regions for each patient were calculated using GRanges (v1.47.0, Source Data)[82], and matched RNAseq log2fold change values were coupled. Cancer gene dependencies for cell lines were downloaded from DepMap (https://depmap.org/portal/, ACH codes used in Supplementary Table 8)[39,83,84], with essential genes and AR-independent differentially expressed genes filtered out. Dependencies for VCaP, LNCaP, and 22Rv1 cell lines were plotted for ARBS with H3K27ac HiChIP interactions with dependent gene promoters, and color-coded for enhancer region and general CNA status in the 101 mPCa patient cohort. Dependent genes were defined with CERES effectivity score < −0.5 and plotted for a selection of dependent genes with all linked ARBS and their occurrence in primary patients. The predominantly occurring copy number alteration for any dependent gene was taken to plot expression for either CNA-affected gene sequences or enhancers, i.e. gains were looked at when such alterations were more prevalent for the gene and enhancer loci.

## General linear model and statistics

Using base R(v4.1.1) function glm, the transcriptional response of patients at H3K27ac HiChIP interacting ARBS-promoter pairs was modeled as a Gaussian distribution determined by a linear combination of ChIP-seq log-transformed MACS score in each patient. Response vector consisted of TMM normalized and matched RNAseq data corresponding to gene of interest. For this, R functions were written to extract ARBS identifiers and ChIP-seq scores from each patient individually, which were combined for each gene in the genome. GLMs were then ran in parallel for every gene ARBS landscape. Interactor $p$-values were extracted from each GLM and plotted for CITED2 with a significance cut-off $p < 0.001$. Linear regression was plotted for a selection of log-transformed ARBS peak scores versus RNA expression for all patients in the primary cohort. Simulated $p$-values were empirically calculated using glmperm package(v1.05) with $n = 1000$ permutations[85].

## scRNA and scATAC-seq data analysis and CICERO accessibility interaction prediction

LNCaP-DMSO scRNA-seq and scATAC-seq data were downloaded[14] and processed using Seurat(v4.0.5)[86] and Signac(v1.4.0)[87]. The scRNA-seq data were filtered to include cells with features >2000, RNA counts <60,000, and mitochondrial reads <20%. The counts were log-transformed and scaled, the 2000 most variable features were selected, principal component analysis was performed, and the top 30 principal components were retained for analysis. The $k$-nearest neighbor and shared nearest-neighbor graphs were constructed at $k = 20$ and cells were clustered with the original Louvain algorithm at a resolution of 0.6.

The scATAC fragments were lifted over from hg38 to hg19 using ucsc-liftover (v366) command-line tools with hg38ToHg19.over.chain file. The peaks were lifted over from hg38 to hg19 using with same chain file and rtracklayer (v1.54.0), then the count matrix was adjusted to include counts for peaks mapping to only one location in hg19. Annotations were obtained from EnsDb.Hsapiens.v75 and the nucleosome signal and TSS enrichment were calculated with Signac. Cells were filtered based on data distribution heuristics to include cells with 2000–50,000 fragments, a fraction of reads in peaks > 0.9, blacklisted regions <0.03%, nucleosome signal <4, and TSS enrichment >1. Differentially abundant peaks were identified by logistic regression using the fragments as latent variables. Significant peaks were identified by an adjusted $p$-value < 0.05 and logFC > 0.1 (Source Data). AR activity was assessed using chromVar(v1.16.0)[88]. Enrichment of ARBS in significant differentially abundant peaks was performed by permutation test using regioneR(v1.26.0)[89] with all peaks as background.

Co-accessibility analysis was performed using Cicero(v1.12.0)[90] and monocle3(v1.1.0)[91,92]. The distance parameter was estimated using a distance constraint of 2.5 Mb with a window size of 5 Mb. Cicero models were constructed, connections assembled, and the cis-co-accessibility networks were generated at a cutoff of 0.1. Midpoints of the connected elements identified by cicero were linked and links with score < 0.05 were filtered out.

## Oligonucleotides

Supplementary Data 1 gives an overview of all used oligonucleotides.

## sgRNA design and production

Guide RNAs for CRISPR/Cas9 deletion was designed to closely flank ARBS of interest and maximize both MIT and Doench scores for optimal guide efficiency during mammalian lentiviral transduction. Guides were annealed and cloned in lentiguide-puro plasmid (Addgene, 52963)[93,94]. All sgRNA lentiguide-puro sequences were Sanger sequences verified using U6-forward primer. gRNA pairs and pools for CRISPR/Cas9 ARBS deletion were pooled equimolarly prior to outgrowth, as confirmed by PCR with U6-F primer and a corresponding reverse sgRNA oligo (Supplementary Data 1, detailed protocol available at https://portals.broadinstitute.org/gpp/public/resources/protocols).

## CRISPR-Cas9 deletion

LNCaP cells were infected with a lentivirus encoding Cas9-eGFP (Addgene, 63592) and GFP-positive cells were flow cytometry selected, after which presence of Cas9 in GFP+ cells was confirmed by western blot with antibodies Cas9 (Cell Signaling #14697, clone 7A9-3A3, diluted 1:1000, positive cell line included) and actin (Sigma Aldrich A2228, clone C4, diluted 1:1000). Cas9 activity was confirmed by lentivirally infecting GFP-Cas9+ cells with a GFP promoter-targeting gRNA pair after which cells were analyzed on a flow cytometer.

sgRNA-containing lentiviruses were produced by transfecting HEK 293T cells ($6.5 \times 10^6$ cells/100 mm plate, 60–70% confluency) with a mix of psPAX2 (Addgene, #12260), pMD2.G (Addgene, #12259), sgRNA containing lentiguide-puro and PEI (PEI:DNA ratio 3:1). Cells were transfected in warm OMEM medium without FCS or PS (Gibco). Medium was supplemented with DMEM (10% FCS, 1% PS) 3 h post-transfection and incubated overnight, after which medium was refreshed with DMEM (10% FCS, 1% PS). Two days post-transfection, the virus-containing medium was harvested by filtering through a 0.45 μm filter and immediately used to infect $2.0 \times 10^5$ GFP-Cas9+ LNCaP cells/six-well plate in RPMI (10% FCS, 1% PS) containing polybrene (10 μg/ml). After two days, cells were selected with puromycin (2 μg/mL). Afterward, the selection was maintained continuously at 1 μg/ml puromycin during culture for 30–40 days until harvest. AR+ breast cancer cell line MDA-MB453 (ATCC HTB-131) was cultured in a similar manner as LNCaP and infected with Cas9-eGFP lentivirus as described above.

## gDNA/RNA isolation, cDNA synthesis for qPCR and PCR verification of CRISPR-Cas9 deletion

Genomic DNA and RNA were isolated from cells washed in ice-cold PBS and resuspended in 0.75 ml TRIzol LS reagent per the manufacturer's instructions (Invitrogen, 15596026). gDNA was precipitated with 100% ethanol, and washed twice with 0.1 M sodium citrate in 10% ethanol, after which gDNA solubilization was facilitated by DNA hydration solution (Qiagen, 158445). RNA was precipitated with isopropanol and the resulting RNA pellet was washed in 75% ethanol and resuspended in nuclease-free water for immediate use in cDNA synthesis or stored at −80 °C for later use.

To generate first-strand cDNA, 2500 ng RNA was primed with 5μM oligo(dT) and 10 mM dNTP mix. cDNA was synthesized as per Super-Script III first-strand synthesis per the manufacturer's instructions

(Invitrogen, 18080-051). cDNA was diluted 1:10 (~20 ng/μl) and used as input in SensiMix SYBR No-ROX kit for qPCR (GC Biotech QT650-20). Primers spanning exon-exon junctions were designed for β-actin and CITED2. The qPCR data were processed using package Rseb[95]. For gDNA verification of successfully removed ARBS, primers were designed to flank Cas9 genome editing sites and used to perform PCR (Thermo-Fisher, F548S) on gDNA and subsequently analyzed by electrophoresis on 1.6% agarose gel.

## CRISPR interference of CITED2 in LNCaP:Suntag-KRAB

The existing Suntag-VP64 construct (pHRdSV40-scFv-GCN4-sfGFP-VP64-GB1-NLS; Addgene, #60904)[42] was adapted by swapping the activating effector VP64 for the repressing KRAB effector using Gibson Assembly and RsrII (Thermo Fisher). Briefly, KRAB was amplified from pLX_311-KRAB-dCas9 (Addgene, #96918) using PCR primers with homology arms that were specifically designed to keep the original scFv-GCN4-sfGFP-effector-GB1-NLS linker architecture intact (Supplementary Data 1). LNCaP cells were subsequently infected with lentivirus containing Suntag-10xGCN4 (pHRdSV40-dCas9-10xGCN4_v4-P2A-BFP, Addgene# 60903). BFP-positive, Suntag-10x positive cells were subsequently sorted in 96-well plates containing LNCaP conditioned medium for monoclonal selection using flow cytometry. After the outgrowth of clones, cells were infected once more with lentivirus containing scFv-GCN4-sfGFP-KRAB and sorted in 96-well plates containing LNCaP conditioned medium for monoclonal selection using flow cytometry.

Resulting grown Suntag10x-KRAB clones were validated using NT and EPCAM targeting pools of sgRNAs. Cells were infected with pools of three sgRNAs, harvested and stained using APC-conjugated CD326 (EPCAM) Monoclonal Antibody (Invitrogen, MA5-38715, EPCAM-APC Clone 323/A3), after which Suntag-KRAB repressing activity was assessed using flow cytometry. Finally, active Suntag-KRAB LNCaP monoclonal cells were infected with sgRNAs targeting CITED2 enhancers, NT targeting sgRNA pool, and CITED2 exon 2 targeting sgRNAs as positive control and grown for 1 week. After harvest, RNA was extracted, cDNA was synthesized and CITED2 qPCR was performed.

## AR enhancer CRISPRi proliferation measurements and HOXB13 ChIP-qPCR

LNCaP:Suntag-KRAB cells were infected with pools of two lentguide puro sgRNAs targeting either the AR promoter (ARp) as a positive control, enhancers AR23 or AR13 and non-targeting NT as negative control and selected using 2 μg/ml puromycin. Cells were grown and variable proliferation speeds were noted between the cell-lines post-infection. After having reached sufficient cell numbers, cells were split and seeded at $2.0 \times 10^3$ cells per well in black 384-well plates as quadruplicates ($n = 4$) in RPMI + 5% DCC + P/S + 0.5 μg/ml puromycin. Proliferation was subsequently monitored by Incucyte ZOOM (Essen Bioscience), which captures microscopic pictures of each well every 4 h. sfGFP expression of LNCaP:Suntag-KRAB cells was used to quantify proliferation. Logistic growth fit was used to model cell line growth, which can be described by the formula $Y = Y_M * Y_0 / ((Y_M - Y_0) * e^{(-k*x)} + Y_0)$, with $Y_0$ as starting population, $Y_M$ as population maximum and $k$ as rate constant.

For HOXB13 binding characterization at the AR13 enhancer, NT and AR13 LNCaP:Suntag-KRAB were grown in biological triplicates ($n = 3$) in a normal medium supplemented with 0.5 μg/ml puromycin to maintain selective pressure until a 15 cm dish confluency of 90% was reached. ChIP was performed[8], but instead, 5 μg HOXB13 antibody (Santa Cruz Biotechnology; sc-66923 clone H-80) was used per condition with 50 μl Protein A Dynabeads (Invitrogen). After ChIP completion, qPCR was performed using aspecific and specific site primers (S1 negative and AR13-q95, respectively) on 1:50 diluted solution from 50 μl of dissolved DNA pellet.

## State-specific sites in ranked ARBS and good/poor outcome predictor

ARBS collections identified in earlier studies spanning PCa disease state transitions[4,5] and association with outcome[8] were downloaded and intersected with ranked ARBS using BEDtools. Primary PCa patient ARBS peaklists were divided in groups of patients with (cases) or without (controls) BCR development. For each group, patients were divided in three groups based on the proportion of good to bad outcome sites in which (1) >1.2 good, (2) 1.2 > mixed > 0.8, (3) poor < 0.8 and used Fisher's exact test for statistical analysis. TARBS and met-ARBS in ranked ARBS were counted for each individual patient inside the case and control groups and plotted in histograms detailing the total ARBS for each group per ranked ARBS occurrence. Statistical difference between histograms was calculated using two-sided Kolmogorov–Smirnov test.

## GO regulatory potential of Met-ARBS, Gene Set Enrichment Analysis, Heatmap clustering

Regulatory Potential scores for genes were calculated using regions peak files for met-ARBS found in 8 or fewer primary patients that developed a BCR recurrence versus those without as input for Cistrome-GO[96] and filtered for genes with RPscore > 0.05. Genes were analyzed for ranked pathway analysis using GSEA(v4.1.0)[97] and MSigDB(v7.3, h.all.v7.0.symbols.gmt Hallmarks)[98]. Met-ARBS genes in 8 or fewer patients were used to filter publicly available and clinically annotated PCa expression data from Taylor[48] and Grasso cohorts[49]. Taylor count data was normalized using DESeq2[99] and differentially expressed genes with FDR < 0.01 were used. Grasso mean $z$-scores for differentially expressed genes were used. Heatmaps for gene expression were plotted using ComplexHeatmap[100] with optimal kclustermeans determined from Silhouette plots generated by R package factoextra (v1.0.7, https://rpkgs.datanovia.com/factoextra/index.html) using Euclidian distance metric.

## Reporting summary

Further information on research design is available in the Nature Portfolio Reporting Summary linked to this article.

## Data availability

All data are available within the article, Supplementary information and Source data file. Source data are provided with this paper. Public datasets are available through GSE accession numbers as reported in Supplementary Table 1 and additional STARR-seq data is deposited at https://www.ncbi.nlm.nih.gov/geo/query/acc.cgi?acc=GSE217319 or accessible through GSEA accession number GSE217319. Source data are provided with this paper.

## Code availability

Code is available through https://github.com/jknp/arbshet.

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

## Acknowledgements

We would like to acknowledge the NKI Genomics Core Facility for Illumina sequencing and bioinformatics support and the NKI Research High-Performance Computing (RHPC) facility for computational infrastructure. We express gratitude to all members of the Zwart and Bergman lab, and members of the NKI Oncogenomics division for helpful scientific discussion. This work was supported by the Prostate Cancer Foundation (21CHAL04), Department of Defense (W81XWH-21-1-0234, W81XWH-19-1-0565), Oncode Institute and Alpe d'HuZes/KWF Dutch Cancer Society (10084).

## Author contributions

J.K., S.E.P.J., A.M.B., and W.Z. conceptualized the study. J.K., P.S. I.P.L.Y., and C.F.H. performed experiments. J.K., T.S., and J.C.S. performed computational analyses. I.P.L.Y. and C.F.H. performed STARR-seq experiments and J.K., T.S., T.M., U.B.A., and N.A.L. performed STARR-seq analyses. I.C., C.J., and R.H. collected and processed tissue samples for pathology. C.G., J.C.S., C.G., J.H.S., S.C.B., E.E., B.P., M.L.F., L.F.A.W., and N.A.L. provided dataset resources. J.K., A.M.B., and W.Z. wrote the original draft manuscript. J.K., T.S., J.C.S., S.E.P.J., I.P.L.Y., C.G., R.H., N.A.L., A.M.B., and W.Z. reviewed and edited the manuscript with input from all co-authors. J.K., A.M.B., and W.Z. supervised the study.

## Competing interests

The authors declare no competing interests.

## Additional information

[1]Division of Oncogenomics, Oncode Institute, Netherlands Cancer Institute, Amsterdam, The Netherlands. [2]Division of Molecular Carcinogenesis, Oncode Institute, Netherlands Cancer Institute, Amsterdam, the Netherlands. [3]Vancouver Prostate Centre, Department of Urologic Science, University of British Columbia, Vancouver, Canada. [4]School of Medicine, Koç University, Istanbul, Turkey. [5]Central RNA Lab, Istituto Italiano di Tecnologia, Genova, Italy. [6]Department of Pathology and Laboratory Medicine, David Geffen School of Medicine, University of California Los Angeles, Los Angeles, USA. [7]The Center for Functional Cancer Epigenetics, Dana-Farber Cancer Institute, Boston, USA. [8]Department of Pathology, Cancer Biology and Epigenetics Group, Portuguese Oncology Institute of Porto and Porto Comprehensive Cancer Center, Porto, Portugal. [9]Department of Physics, Simon Fraser University, Burnaby, Canada. [10]Department of Medical Oncology, The Center for Functional Cancer Epigenetics, Dana Farber Cancer Institute, Boston, USA. [11]Koç University Research Centre for Translational Medicine (KUTTAM), Koç University, Istanbul, Turkey. [12]Division of Medical Oncology, Netherlands Cancer Institute, Amsterdam, the Netherlands. [13]Laboratory of Chemical Biology and Institute for Complex Molecular Systems, Department of Biomedical Engineering, Eindhoven University of Technology, Eindhoven, The Netherlands. ✉e-mail: a.bergman@nki.nl; w.zwart@nki.nl

