## [Peer Review File · Nature Communications]

Extensive androgen receptor enhancer heterogeneity in primary prostate cancers underlies transcriptional diversity and metastatic potentialREVIEWER COMMENTS

Reviewer #1 (Remarks to the Author): Expert in prostate cancer genomics and epigenomics, and computational genomics

Authors present an impressive analysis on the heterogeneity of AR binding sites in prostate cancer patient tissues. This analysis builds on data from their earlier characterization of AR regulome subtypes. Current study highlights a large fraction of AR binding sites that are present only in a single tumor or a subset of studied tumors. By integration with multiple published datasets and data types extensive analysis is presented to address the functional significance of these binding sites. In addition, new experimental data from STARR-seq assay as well as from CRISPR-Cas9 mediated genome editing is presented to validate function of selected binding sites. Overall this is an interesting and well executed study. Presented analysis and data is of high quality.

Specific comments:

In comparison to healthy tissues, confounding issue is much lower fraction of AR positive luminal cells within tissues than with in tumors foci. How this been addressed in the quality control? What is the impact of presumably lower fraction of AR positive cells in normal tissue and/or variable fraction of AR positive cells in tumors?

While authors clearly justify that identified sample specific peaks are of high quality, have the missing peak calls been addressed? That is, to confirm that peak is infact sample specific due to the lack of signal in other samples, not just due to missing peak call. Did you quantify the signal at the peak locations across all samples? That is, is there signal above background in some samples that was missed by peak calling? If peaks are quantified (vs just called by peak caller) across samples, can similar fractions of SH, PS, and UN peaks still be observed?

Presentation of new STARR-seq data is not clear. From figure 2 it is hard to see how the data supports authors view that heterogenous AR binding is functional (vs. only previously reported tissues specific sites with high recurrence). I assume key for interpretation are the newly designed STARR-seq ARBS sites, which in currently figure are not clearly presents i.e. not easy to see of these distribute evenly throughout SH, PS, and UN peaks. More clear visualization preferably with statistics to support the distribution of peaks would be beneficial.

CNV analysis (in 3B) does not seem surprising as large part of the genome is aberrated in aggressive cancers. Can this analysis be extended to justify the CNV regulation be highlighting the ARBS with focal copy number alterations or rearrangements? That is, is there evidence that ARBS would actually be targeted by CNVs (beyond the known AR enhancer amplification)?

GLM models are used to link ARBS to target gene expression. While p-values are provided more details needs to be reported. Does the model assumptions hold and how accurate are the predictions obtained? Are the MACS values robust (normalized) across samples? One option would be to use data permutation to obtain and report empirical p-values.

As authors make the case that "heterogenous ARBS contribute to progression", from the perspective of the progression does this mean that only the samples that have these ABRS progress, and thus is the observed heterogeneity only due to fact that samples considered are primary tumors of which majority presumably does not have metastatic/progression potential at the time of the surgery. Thus, would be good to clarify if authors claim is that trajectories to progression are heterogenous, if there is a specific (recurrent) signature for majority of the progressed tumors (thus progressed tumors would be less heterogenous), or if the argument of heterogeneity is relevant only in the context of the sample set studied.

Reviewer #2 (Remarks to the Author): Expert in cancer epigenomics, functional genomics and ChIP-seq

In this manuscript, Kneppers et al. investigated androgen receptor enhancer heterogeneity in prostate cancer. They find high levels of heterogeneity, which may lead to differences in gene expression and clinical potential. Overall the manuscript is well written, and the experiments are well performed.

However, I have several questions about the manuscript:

1. To what extent is the heterogeneity observed because of technical or methodological issues instead of being actual biological variation between the samples? I understand the authors performed QC testing to verify that the peaks called are real, but there might be different levels of real peaks identified in different libraries. It would be interesting to know what is the level of variation observed if technical replicates were to be performed on the same samples.
2. Do differences in sequencing depth of libraries affect the samples?
3. What sorts of quality controls are performed on the samples to ensure that the heterogeneity observed is not because of, for example, dying cells in the sample?
4. The statement in the introduction that "These findings indicate that epigenetic heterogeneity in primary disease is directly informative for metastatic potential." Suggests that the authors found that epigenetic heterogeneity could predict metastatic potential. However, the authors did not look at primary samples that progressed on to metastasis. I understand this type of analysis is very difficult to do because of sample availabilities, but I think this sentence should be toned down.
5. The authors performed a CRISPR enhancer knockout analysis and found this affected transcriptional potential. Does it affect anything else besides transcription? Are tumors grown from cells with the enhancer knockout smaller? How about metastatic potential? Does this change?
6. Continuing along the lines of the CRISPR enhancer knockout, if the authors were to look at one cell line that has the enhancer and another cell line that does not have the enhancer, and do the knockout in both cell lines, would the cell line without the enhancer originally also show any changes in transcriptional expression? This would suggest that it is the enhancer that leads to the difference in expression levels and not simply the presence of the DNA sequence.
7. One of the treatments for prostate cancer is androgen receptor blockers. How does enhancer heterogeneity affect the functioning of androgen receptor blockers? How about other drugs, such as CDK7 inhibitors or other inhibitors that affect superenhancers – what effect does enhancer heterogeneity have on how these drugs work?
8. What could be the likely causes of AR heterogeneity in cancers?

Reviewer #3 (Remarks to the Author): Expert in prostate cancer functional genomics and therapy

In this study, Kneppers, et al investigated the extent and consequences of inter-tumor heterogeneity of AR enhancer selectivity. Through analyzing the ARBS in a primary prostate cancers cohort, they revealed that the AR enhancer usage is highly heterogeneous in primary PCa. they found that ARBS are affected by SNVs and CNVs acquired during PCa progression. Furthermore, they characterized the biological consequences and clinical implications of such heterogeneity in PCa progression and metastasis. Through STARR-seq, they confirmed that the ranked ARBS exposes hierarchical enhancer activity. Through modelling H3K27ac-based HiChIP interacting ARBS-promoter pairs and matched gene expression from 88 primary patients in a generalized linear model (GLM), they predicted which ARBS influences gene expression most and found that the less-frequently shared ARBS can have the largest impact on transcriptional output. Finally, they investigated whether heterogeneous ARBS play a role in PCa progression to metastatic disease and observed that met-ARBS were selectively enriched in partial shared and unique ARBS in patients whose tumors ultimately progressed.

Overall, this is an interesting work and may have potential high impact for the prostate cancer research fields. However, I do have several questions about the manuscript:

1. Line 77, why would the authors set the cutoff of shared or partially shared or unique peak at 68% or

2% percent of patients? any particular reason for this selection of statistic reasoning?

2. Line 81 and Fig1b and c, it seems that there are significantly difference between the SH ARBS between the LNCaPBR, 22Rv cells vs the LNCaP and VCaP cells, which may suggest those SH ARBS contribute to ADT resistance? what about PS and UN ARBS between those lines? authors should also test some other enzalutamide resistant LNCaP sub lines such as LNCaP-95 or C4-2.

3. same Fig1c, is there significant more SH sites in MDAMB453 compared to LSHAR? looks like there is a clear difference, what would be the interpretation?

4. Fig2e, it seems that mutation rate of SNVs is decreasing across all ARBS from primary to metastatic samples? isn't this controversial to the accumulation of SNVs during PCa progression?

5. Fig3c, for all of the 25 essential genes, is there difference between their association with PS or UN sites? for example, UCHL5 interacts with significant number of PS sites, while ZBTB10 interacts with UN sites much more. what would be the interpretation of those differences between the 25 genes?

6. Fig4e shows the guides pair 12 is the most effective ones to inhibit CITED2, but the effect of other guide pairs is not shown (only the pooled one is shown). What about the other guide RNAs targeting the UN ARBS? Also, since these pairs of guide RNAs targeting the whole sites, it is necessary to perform rescue experiment to confirm the biological consequence is real due to the deletion of these sites.

7. Fig4F, G, what are the gene motifs enriched for AR13 sites beside HOXB13? Do HOXB13 motifs also enriched in other AR2 or AR23 sites? more importantly, the author should show there is indeed downregulation of HOXB13 signature in those AR13 KD cells, and perform rescue experiment rescue the cell proliferation.

8. Fig4F, G, the authors didn't explain why AR13, 23 was chosen here, vs the ARBS shared by 54 patients in previous analysis? what is the rationale of choosing all different cutoffs of shared ARBS throughout the paper?

9. Fig5B, how many poor vs good outcome ARBS are SH vs PS vs UN sites? Are there enrichment of poor outcome ARBS for the UN sites compared to SH sites?

10. Fig5D, what about the met vs primary ARBS in the SH ARBS? no difference? is there enrichment of those met-ARBS, especially the ones at UN-ARBS, enriched in any genes known to promotes PCa metastasis?

11. For the STARR-seq data, it's better to validate specific inducible/constitutive/inactive STARR-seq enhancers in vitro using luciferase reporter assay.

Reviewer #4 (Remarks to the Author): Expert in prostate cancer epigenomics

Kneppers et al performed analyses using publicly available data (generated from Zwart lab + other researchers) to study androgen receptor binding sites in primary prostate cancer patients. Many useful integrative analyses are done, but it is not clear what is the key take-home/novel message of this manuscript. Researchers already know that there are heterogeneous epigenetic signatures among prostate tumor samples. See below to find specific comments/questions of each analysis/figure.

Figure 1: Averaging 5,787 peaks per tumor seems to be too low and the number of identified peaks per tumor seems to vary a lot. Detailed information of identified peaks per tumor is needed as technical variation can lead to those variations. It is not clear if the author evaluated quality of each dataset (e.g. use ENCODE guideline to measure NSC, RSC to check enrichment level, read

numbers) besides checking GC contents. Provide how many peaks are found from each sample, how many are categorized to SH/PS/UN per tumor. Current suppl table/figures do not include those.

Describe whether peak height (ChIP-seq signal) is considered to rank peaks besides overlapping identified peaks among datasets (SH/PS/UN). False positive peaks may exist in unique ones.

Figure 1D and reference 7: Authors stated that AR enhancer heterogeneity is not tumor-intrinsic, but instead patient-intrinsic. Generate Figure 1D-like figure using cancer-specific/normal-specific ARBS only to test that. Explain how patient-intrinsic ARBS can lead to transcriptome changes linked to cancer subtype/metastatic cancer.

Figure 1H, Suppl Fig 1: Motif analysis - Authors indicated that MED1 and RNAPII subunits were enriched in UN-ARBS. How many peaks have each motif? Are these located in promoter/enhancer? Enrichment score is not informative if the number of peaks with motifs is too small. Provide more information and interpret data more thoroughly.

Figure 1H: Letters are too small and it's not informative. Either replace with a barplot or move to supplemental figures.

Figure 2A: Not clear if machine learning is appropriate to analyze patient AR ChIP-seq data for a massive parallel reporter assay data generated in one cell line. As Figure 1 showed, the number of ARBS and ARBS differ among patients.

Authors performed additional STARR-seq targeting at 2,495 randomly sampled heterogeneous ARBS, which is good. However, more information needs to be provided. How many were found in shared/PS/unique and how many are found in ind/cons/inactive enhancers? Provide those in suppl data. What is the take-home message? Unique ones are less functional?

Figure 2D-F: Are rSNP/SNV alleles from primary AR ChIP-seq data used to evaluate rSNP/SNV allele or just genomic location of reported rSNP/SNV regions were overlapped to ARBS? It would have been better if authors could have done more analyses using allele information from ChIP-seq data.

Figure 3A: Promoter-enhancer interaction data used from this study is from one cell line while AR ChIP-seq data is from many patients (primary tumor samples).

Figure 3B: Are large CNA gains/losses tested in AR ChIP-seq data? Just genomic location of reported CNA regions were overlapped to ARBS? It would have been better if authors could have done checked CNA using ChIP-seq/RNA-seq data and show CNA relationship to ARBS using the datasets from the same sample.

Figure 4 and Suppl Figure 4: Figure 4C and D are done in what cell line? Both LNCaP and 22Rv1 or just one cell line? Are CRISPR/Cas9 experiments done in pool or single cells are cloned and performed RT-qPCR? It looks like that it's in the pool, right? If that's the case, H3K27ac ChIP is done to check the deletion effect in the pool? What controls are included? Explain more in detail.

Figure 5. TARBS and met-ARBS analysis finding (e.g. lines 418-422) is contradictory to the recent manuscript ref 7? Explain. It is not clear how transcription levels of ARBS' target genes are studied. Are HiChIP data used to identify target genes or nearby genes are used? Explain. Provide a list of identified genes in suppl table.

Reviewer comments with authors' replies

Reviewer #1 (Remarks to the Author): Expert in prostate cancer genomics and epigenomics, and computational genomics

Authors present an impressive analysis on the heterogeneity of AR binding sites in prostate cancer patient tissues. This analysis builds on data from their earlier characterization of AR regulome subtypes. Current study highlights a large fraction of AR binding sites that are present only in a single tumor or a subset of studied tumors. By integration with multiple published datasets and data types extensive analysis is presented to address the functional significance of these binding sites. In addition, new experimental data from STARR-seq assay as well as from CRISPR-Cas9 mediated genome editing is presented to validate function of selected binding sites. Overall this is an interesting and well executed study. Presented analysis and data is of high quality.

We thank the reviewer for their time, effort and highly constructive suggestions and feedback in critically evaluating our manuscript. We greatly appreciate the reviewer's kind comments on the data and analysis.

Specific comments:

In comparison to healthy tissues, confounding issue is much lower fraction of AR positive luminal cells within tissues than with in tumors foci. How this been addressed in the quality control? What is the impact of presumably lower fraction of AR positive cells in normal tissue and/or variable fraction of AR positive cells in tumors?

We agree with the reviewer that variations in AR+ cells in tumor and normal tissue may confound comparisons. Of note, as the original manuscript was submitted as 'analysis paper' to another journal in the Nature press family, many of the -omics datasets used in this manuscript represent a new analysis of existing datasets reported previously by us and others.

Regarding the normal versus tumor ChIP-seq datasets: These were reported previously in (Ref. 4 Pomerantz et al., 2015 Nat. Genet., Ref. 5 Pomerantz et al., 2020 Nat. Genet. and Ref. 34 Mazrooei et al., 2019 Cancer Cell). For normal tissue, epithelial cell-rich areas were selected by an expert pathologist, and for the dataset described in Ref. 5, AR immunohistochemistry has been performed for both healthy and tumor tissue, as shown below and reported in Ref. 5. Between these, AR positivity was largely comparable for healthy and tumor samples, with high expression levels for all samples analyzed (~70-90%). To further address this reviewer comment, we specifically analyzed RNA-seq AR transcript levels and observed no correlation between AR transcript levels and the number of AR peaks, indicating that the samples we analyzed in this study had AR levels that were not a rate-limiting factor in the number of AR peaks in the tissue sample (Supplementary Figure 1F). Based on these observations, we have updated the results section accordingly.

Subject	Tumor enrichment in ChIP-seq sample (%)	AR		Gleason Score
		Normal prostate (%)	Prostate tumor (%)	
NKI_1	60	90	80	4+5=9
NKI_5	65	80	90	3+4=7
NKI_7	70	90	90	3+4=7

NKI_13	30	90	90	3+4=7
NKI_20	70	90	70	3+4=7
NKI_23	40	80	90	3+4=7
NKI_27	30	90	90	4+3=7
NKI_29	40	90	80	4+4=8

While authors clearly justify that identified sample specific peaks are of high quality, have the missing peak calls been addressed? That is, to confirm that peak is infact sample specific due to the lack of signal in other samples, not just due to missing peak call. Did you quantify the signal at the peak locations across all samples? That is, is there signal above background in some samples that was missed by peak calling? If peaks are quantified (vs just called by peak caller) across samples, can similar fractions of SH, PS, and UN peaks still be observed?

The reviewer raises an important question and suggests an appropriate testing method to account for false negatives and determine if peaks, when quantified instead of called, would give a different result. These analyses are presented in Supplementary Figure 1H. Here, we display the signal intensity at all called peaks in the sample, in comparison to the signal at regions not called in this specific sample (but that were called in other samples). Based on this analysis, we conclude that patient-specific ARBS are indeed called correctly due to lack of signal at regions not called in this sample, but that have been called in other samples. These analyses illustrate that there is little signal above background which could cause false negatives, which would lead to a higher number of UN-ARBS versus PS-ARBS, minimizing the chance of false negatives (next to false positive analysis as presented in Supplementary Figure 1B).

Finally, we would like to underline that even without such validation analyses, UN-ARBS which should have been PS-ARBS due to false negatives (so consequently ARBS in 2 or maximally 3 patients) would not have significantly affected the clinical conclusion drawn from the data presented in Figure 5. We have updated the discussion section to better reflect this consideration.

Presentation of new STARR-seq data is not clear. From figure 2 it is hard to see how the data supports authors view that heterogenous AR binding is functional (vs. only previously reported tissues specific sites with high recurrence). I assume key for interpretation are the newly designed STARR-seq ARBS sites, which in currently figure are not clearly presents i.e. not easy to see of these distribute evenly throughout SH, PS, and UN peaks. More clear visualization preferably with statistics to support the distribution of peaks would be beneficial.

We apologize for the unclear representation of these data, and thank the reviewer for highlighting this issue. We agree with the reviewer's concern and have added the new Supplementary Figure 2A,B to address this, which more clearly visualizes the distribution, amount and hierarchy of the extra STARR-seq sites that are active in this LNCaP context, while some of these extra sites show activity in heterogeneous PS- and UN-ARBS. Additionally, to further validate the activity readout of the STARR-seq data, newly generated luciferase reporter assays at 14 regions were included, confirming our original results and observations through an orthogonal approach. These data are presented in the new Supplementary Figure 2C,D and are discussed in the results section.

CNV analysis (in 3B) does not seem surprising as large part of the genome is aberrated in aggressive cancers. Can this analysis be extended to justify the CNV regulation be highlighting the ARBS with focal

copy number alterations or rearrangements? That is, is there evidence that ARBS would actually be targeted by CNVs (beyond the known AR enhancer amplification)?

While the intended message of these analyses was aimed to determine CNVs in relation to the hierarchy of AR peaks relative to inter-sample heterogeneity, we fully agree with the reviewer that the current analysis should be expanded. Following the reviewers' advice, we have updated our CNV analysis in Figure 3B to include smaller SVs that have been observed in metastatic prostate cancer patients (Ref., 11 Quigley et al. 2018 Cell). Intriguingly, we hardly find any breakpoints nor insertions to occur directly at ARBS in comparison to deletions, inversions and tandem duplications (Supplementary Table 12). Interestingly, especially deletions occur frequently at well-described PCa SV loci, with chr2q (SPOPL), chr18q (DCC/BCL2) and chr11 being highly affected, but this may not be purely due to regulatory element presence.

Of note, in metastatic prostate cancer we previously identified a clear enrichment of CNVs at enhancer elements driving expression of critical AR drivers, including MYC, FOXA1, HOXB13, NKX3-1 and indeed AR (Ref. 5 Pomerantz et al., 2020 Nat. Genet. and Ref. 11 Quigley et al., 2018 Cell). These results provide evidence that specific ARBS are targeted by CNVs in contexts such as mCRPC. This argumentation has now been added to the results section.

GLM models are used to link ARBS to target gene expression. While p-values are provided more details needs to be reported. Does the model assumptions hold and how accurate are the predictions obtained? Are the MACS values robust (normalized) across samples? One option would be to use data permutation to obtain and report empirical p-values.

To clarify our GLM model, we have updated the main text to provide a better explanation and present the suggested model assumption tests which are now included in Supplementary Figure 4A. These tests show that the model assumptions hold and include a normal Q-Q plot which shows linearity of the points suggesting that the data are normally distributed, while the Residuals vs Leverage plot shows the relative influence of each patient on model predictions, which all fall within a Cook's distance of 0.5. This indicates that individual patients do not significantly distort model predictions or influence prediction accuracy. Moreover, following the reviewer's advice, we have now executed the proposed data permutation to test for model robustness, yielding a comparable empirical model p-value ($p_{\text{obs}} = 0.0316$, $p_{\text{sim}} = 0.035$). We thank the reviewer for highlighting these relevant issues, which helped us to position these data better.

As authors make the case that "heterogeneous ARBS contribute to progression", from the perspective of the progression does this mean that only the samples that have these ARBS progress, and thus is the observed heterogeneity only due to fact that samples considered are primary tumors of which majority presumably does not have metastatic/progression potential at the time of the surgery. Thus, would be good to clarify if authors claim is that trajectories to progression are heterogeneous, if there is a specific (recurrent) signature for majority of the progressed tumors (thus progressed tumors would be less heterogeneous), or if the argument of heterogeneity is relevant only in the context of the sample set studied.

The reviewer raises a very interesting point. Of note, our study is based on a matched case-control cohort in which 50% of the cases progressed, with the most-heterogeneous ARBS only in the patients that did progress being enriched for met-ARBS (ARBS that demarcate metastatic cancer, as a common denominator in these tumors) (Figure 5, Ref. 3 Stelloo et al., 2018 Nat. Commun. from which Supplementary Figure 1 is shown below for quick reference). Based on this design, we have a relatively

large patient population progressing -as assessed on biochemical relapse- as compared to those that don't.

However, as we show now that the most-heterogeneous ARBS are associated with metastasis-specific ARBS and biochemical relapse, these data support a conclusion that the trajectories to progression are heterogeneous, in which a wide spectrum of ARBS can drive disease progression. Upon disease progression, these more-heterogeneous sites are enriched for those tumors that indeed progress, manifesting as met-ARBS. As potential signatures/pathways driving this phenotype, we included geneset enrichment analyses in the manuscript (Figure 5F), revealing differential expression of these genes specifically in metastases as compared to primary tumors (Figure 5D,E), with enrichment of pathways that are known to be involved in prostate cancer progression, including cholesterol synthesis, mTORC1 signaling, androgen response and WNT beta-catenin signaling (Figure 5F).

We have extended the discussion section of the manuscript to highlight this critical issue more clearly, and we thank the reviewer for raising this highly relevant point.

Reviewer #2 (Remarks to the Author): Expert in cancer epigenomics, functional genomics and CHIP-seq

In this manuscript, Kneppers et al. investigated androgen receptor enhancer heterogeneity in prostate cancer. They find high levels of heterogeneity, which may lead to differences in gene expression and clinical potential. Overall the manuscript is well written, and the experiments are well performed. However, I have several questions about the manuscript:

We thank the reviewer for critical evaluation of the manuscript and for formulating many helpful comments, which prompted us to perform additional experiments to strengthen the manuscript. Moreover, we are highly appreciative of the kind comments on the manuscript and experiments.

1. To what extent is the heterogeneity observed because of technical or methodological issues instead of being actual biological variation between the samples? I understand the authors performed QC testing to verify that the peaks called are real, but there might be different levels of real peaks identified in different libraries. It would be interesting to know what is the level of variation observed if technical replicates were to be performed on the same samples.

The reviewer raises a highly relevant issue, in relation to sample QC testing. Indeed, we have performed extensive QC analyses on all the data used in this manuscript, as well as in the original story which

described most of the raw data used for this study (Ref. 3 Stelloo et al., 2018 Nat. Commun.). On top of the originally performed QC, we have now added Supplementary Tables 2-4 detailing tumor cell percentage, peak distribution and read QC data including NSC, RSC and FRiP scores. Moreover, we have updated Supplementary Figure 1 extensively to reflect these QC measures and included Supplementary Figure 1E to visualize peak distribution. As shown previously (Ref. 8 Stelloo et al., EMBO Mol. Med. 2015), the technical variation between subsequent sections from the same tumor is limited, with a strong correlation between them (87% overlap of peaks; R=0.76).

More recently, we reported that even between metastases in different organs from within the same patient, strong overlap of AR peaks was observed (Ref. 7 Severson et al., 2021. Mol Oncol). These data suggest that a) inter-sample variations as described in this manuscript are not due to technical limitations b) variations are not inter-sample centric, but rather inter-patient centric.

To further address these concerns, we show additional QC analyses from (Ref. 5 Pomerantz et al., 2020 Nat. Genet.) on the tumor samples that were analyzed, including additional AR immunohistochemistry data showing a high and comparable percentage of cells expressing AR, both in healthy prostate epithelial cells and tumor cells (shown below).

Subject	Tumor enrichment in ChIP-seq sample (%)	AR		Gleason Score
		Normal prostate (%)	Prostate tumor (%)	
NKI_1	60	90	80	4+5=9
NKI_5	65	80	90	3+4=7
NKI_7	70	90	90	3+4=7
NKI_13	30	90	90	3+4=7
NKI_20	70	90	70	3+4=7
NKI_23	40	80	90	3+4=7
NKI_27	30	90	90	4+3=7
NKI_29	40	90	80	4+4=8

Furthermore, we analyzed RNA-seq data from all 100 tumor samples in the primary tumor cohort for AR expression, which was tested for possible correlation with the number of identified AR ChIP-seq peaks (new Supplementary Figure 1F). In these analyses, no correlation was observed between AR transcript levels and the number of peaks, suggesting that AR levels do not represent a rate-limiting factor on the number of AR peaks identified.

2. Do differences in sequencing depth of libraries affect the samples?

Indeed, variations in sequencing depth are present in these samples which could theoretically affect the samples, among others on the level of number of AR peaks. The read numbers and number-reads-in-peaks were originally listed in (Ref. 3 Stelloo et al., 2018 Nature Communications), but included again for clarity -albeit in a different visualization- in this manuscript in Supplementary Table 3 and Supplementary Fig. 1F. As additional new QC analysis, we tested correlation between the number of peaks versus sequencing depth, which is now included as a new Supplementary Figure 1F. Here, no correlation was observed, indicating that the sequencing depth did not impact our analysis on the level of AR peak numbers.

3. What sorts of quality controls are performed on the samples to ensure that the heterogeneity observed is not because of, for example, dying cells in the sample?

We thank the reviewer for highlighting this relevant point. Sample collection was performed in a highly standardized and optimized setup at the Department of Pathology of Portuguese Oncology Institute of Porto. After surgery, all radical prostatectomy samples were directly processed by the same dedicated uropathologist (prof. dr. Rui Henrique) in a standardized manner. In brief, the whole prostate was cut transversally to the main axis into slices of approximately 6-7mm thickness. Each slice was subsequently cut into quadrants and the fragment of each quadrant was halved, producing twin fragments, of which one was immediately snap-frozen and the other immediately placed in a cassette and immersed in neutral buffered formalin after which it was processed for paraffin embedding and routine histopathological assessment, as well as immunohistochemistry. This procedure minimized cell death and autolysis due to technical work-up, which is validated on H&E staining, as no artifacts of poor fixation were disclosed. Moreover, the fragments (which always contain normal prostate tissue) were stained for standard neuroendocrine markers Chromogranin A and Synaptophysin, showing low expression levels in normal epithelium (approximately 1%), as expected and in line with published literature (Butler et al., 2021 *Precis Clin Med*). Synaptophysin immunostaining also highlighted the nerve fibers present in the prostatic stroma. These statements have now been included in the results and methods section.

Furthermore, samples were stained for tumor cell percentage, which was determined by the expert pathologist (RH) and low-tumor cell containing samples were enriched through macro-dissection prior to cryo-sectioning to avoid significant tissue heterogeneity. Information on tumor cell percentage has now been included in Supplementary Table 2.

4. The statement in the introduction that “These findings indicate that epigenetic heterogeneity in primary disease is directly informative for metastatic potential.” Suggests that the authors found that epigenetic heterogeneity could predict metastatic potential. However, the authors did not look at primary samples that progressed on to metastasis. I understand this type of analysis is very difficult to do because of sample availabilities, but I think this sentence should be toned down.

We thank the reviewer for mentioning this issue and we fully agree. Indeed, we did not look at primary samples that progressed on to metastasis, but we did look at primary samples in patients that did or did not develop a biochemical recurrence. Therefore, we now reformulated the sentence, staying closer to the actual data and remaining concurrent with our observations: “These findings indicate that epigenetic heterogeneity in primary disease is directly informative for biochemical relapse.”

5. The authors performed a CRISPR enhancer knockout analysis and found this affected transcriptional potential. Does it affect anything else besides transcription? Are tumors grown from cells with the enhancer knockout smaller? How about metastatic potential? Does this change?

The reviewer suggests important measurements to comprehensively assess oncogenic potential besides transcription, but KOs can be technically challenging when cell lines are dependent on the gene-of-interest for growth. CRISPR inhibition through guiding catalytically inactive dCas9 fused to a repressing Krüppel Associated Box (KRAB) domain is conceptually well-suited for repressing enhancers over knocking such enhancers out, but has given us variable levels of repression in LNCaP cells. Therefore, we opted for a different CRISPR inhibition approach through adapting the Suntag system (Ref. 43 Tanenbaum et al., 2014

Cell), which recruits up to 10 KRAB effectors to a single locus for more robust CRISPR inhibition at enhancers than conventional dCas9-KRAB CRISPRi (Supplementary Figure 6A,C).

As main advantage of this strategy, CRISPRi regionally suppresses loci without the need for a classical NGG PAM directly at AR motif elements for successful perturbation. Similarly, this approach was used for perturbing ARBS connected to the CITED2 locus (Supplementary 6F), which shows varying degrees of CITED2 mRNA level knockdown for different enhancers. No effect on cellular fitness, proliferation speed nor tumor colony size was observed upon CITED2 enhancer suppression, which is in agreement with publicly available gene CRISPR KO screens (Ref. 40 Tsherniak et al., 2017 Cell) in prostate cancer cell lines.

6. Continuing along the lines of the CRISPR enhancer knockout, if the authors were to look at one cell line that has the enhancer and another cell line that does not have the enhancer, and do the knockout in both cell lines, would the cell line without the enhancer originally also show any changes in transcriptional expression? This would suggest that it is the enhancer that leads to the difference in expression levels and not simply the presence of the DNA sequence.

We thank the reviewer for raising this highly interesting question. To address this point, we expanded our analyses to another AR+ cell line model: the AR-dependent breast cancer cell line MDA-MB453, which lacks an ARBS at the CITED2 ARBS CI54, in contrast to LNCaP (Supplementary Figure 6G). We generated and validated a stable MDA-MB453 Cas9 cell line (Supplementary Figure 6A) and performed efficacious enhancer KO experiments alongside LNCaP Cas9 as described previously (Supplementary Figure 6H). We observe that perturbation of CI54 in MDA-MB453 does not decrease CITED2 expression compared to NT, while perturbation of CI54 in LNCaP leads to the previously reported CITED2 transcriptional decrease. These results confirm the original interpretation of our findings, and illustrate that this particular ARBS (CI54) has impact on CITED2 mRNA levels when active. We have added this conclusion to the results section.

7. One of the treatments for prostate cancer is androgen receptor blockers. How does enhancer heterogeneity affect the functioning of androgen receptor blockers? How about other drugs, such as CDK7 inhibitors or other inhibitors that affect superenhancers – what effect does enhancer heterogeneity have on how these drugs work?

The reviewer raises an interesting clinical discussion. As the met-ARBS identified in Ref. 5 Pomerantz et al., 2019. Nature Genetics represent ARBS that are selectively enriched in castration-resistant metastatic lesions, and these met-ARBS were enriched specifically in heterogeneous ARBS observed in tumors from patients who progressed later in life (Figure 5), the more-heterogeneous ARBS may have direct clinical implications in relation to response to AR targeted therapy. Along the same lines, the 339 ARBS we identified previously (Ref. 8 Stelloo et al., 2015 EMBO Mol. Med.) to separate primary prostate cancers from heavily pretreated progressive samples, were also differentially enriched in their heterogeneity among patients in our primary treatment naïve prostatectomy cohort (Figure 5A,B).

Jointly, these data support the conclusion that AR sites that are more-heterogeneously distributed among patients are associated with biochemical progression (Ref. 3 Stelloo et al. 2018 Nat. Commun.) as well as response to AR-blocking therapeutics (Ref. 8 Stelloo et al., 2015 EMBO Mol. Med.). Of note, as cell lines do not represent the spectrum of clinical heterogeneity, clinical cistromic datasets are needed to sufficiently address this point. Along these lines, since no clinical AR cistromic datasets are available at this stage regarding CDK7 inhibitors or other inhibitors that affect superenhancers, it is currently not

possible to test this highly interesting hypothesis. We have updated the discussion section accordingly to address this point and highlight the study of these newer therapeutic interventions as an interesting future field-of-focus once such cistromic datasets are available.

8. What could be the likely causes of AR heterogeneity in cancers?

This is a very interesting and important question. We recently wrote an extensive review (Kneppers et al., 2022 Springer Nature Books, Nuclear Hormone Receptors in Human Health and Disease, *in press*) on the topic of epigenetic heterogeneity in prostate cancer which addresses the current knowledge on this complex question. To date, numerous factors have been reported that impact AR cistromic profiles, including but not limited to: 1) coding somatic variants such as TMPRSS2-ERG fusions, 2) non-coding somatic variants such as AR enhancer overexpression in metastatic patients (Ref. 11 Quigley et al., 2018 Cell; Ref. 13 Takeda et al., 2018 Cell), 3) large scale structural variation in metastatic prostate cancer like chromosome 8p deletions and 8q gains resulting in loss of LPL and amplification of oncogene MYC, 4) pioneer factor mutants which frequently manifest as FOXA1 mutations that alter pioneering activity and perturb normal luminal epithelial differentiation programs, 5) abnormal co-opting of pioneer factors with other pioneer factors like HOXB13, 6) Different AR cistromes associated with different cellular contexts and tumor micro-environments, 7) clonal heterogeneity that arises from prostate cancer primary tumor multifocality, 8) induction of therapy resistance and AR cistromic heterogeneity through strong AR signaling blockade with inhibitors such as enzalutamide, 9) germline risk single nucleotide polymorphisms in AR enhancers.

The combined output of these (epi)genetic features likely underlies the clinical heterogeneity of AR enhancers that we report here, which represents a major focus point of our lab for the near future. We now expanded the discussion section to highlight specific and known parameters that cause AR enhancer heterogeneity and pinpoint this as future field-of-focus.

Reviewer #3 (Remarks to the Author): Expert in prostate cancer functional genomics and therapy

In this study, Kneppers, et al investigated the extent and consequences of inter-tumor heterogeneity of AR enhancer selectivity. Through analyzing the ARBS in a primary prostate cancers cohort, they revealed that the AR enhancer usage is highly heterogeneous in primary PCa.

they found that ARBS are affected by SNVs and CNVs acquired during PCa progression. Furthermore, they characterized the biological consequences and clinical implications of such heterogeneity in PCa progression and metastasis. Through STARR-seq, they confirmed that the ranked ARBS exposes hierarchical enhancer activity. Through modelling H3K27ac-based HiChIP interacting ARBS-promoter pairs and matched gene expression from 88 primary patients in a generalized linear model (GLM), they predicted which ARBS influences gene expression most and found that the less-frequently shared ARBS can have the largest impact on transcriptional output. Finally, they investigated whether heterogeneous ARBS play a role in PCa progression to metastatic disease and observed that met-ARBS were selectively enriched in partial shared and unique ARBS in patients whose tumors ultimately progressed.

Overall, this is an interesting work and may have potential high impact for the prostate cancer research fields. However, I do have several questions about the manuscript:

We thank the reviewer for the highly constructive and helpful comments and suggestions, which we have implemented and have significantly strengthened the manuscript. Moreover, we appreciate the reviewer's kind remarks regarding the work.

1. Line 77, why would the authors set the cutoff of shared or partially shared or unique peak at 68% or 2% percent of patients? any particular reason for this selection of statistic reasoning?

We thank the reviewer for raising this important issue regarding the selection of ARBS for the SH, PS and UN ARBS bins. First of all, UN-ARBS are by definition unique to a single patient and are therefore limited to 1/88 patients of the entire cohort. Consensus building usually looks at sites shared in at least half of the patients (44/88), but we chose a more stringent cutoff that reflects $60/88 = 0.68$, which encapsulates 2 standard deviations on a normal distribution, which we reasonably argued would capture most commonalities between patients.

However, to further address this concern and demonstrate that the conclusions following from these cutoffs are robust, we recalculated p-values by setting the SH definition more leniently to 44-88, with PS 2-43 patients and UN 1 patient. In that case, we find comparably low p-values ($p\text{-value} < 2e^{-16}$) of SH enrichment when calculating cell-line SH enrichment with these cutoffs, which holds also true for SNV mutation data.

Most importantly, regarding the cutoffs and enrichment analysis in Figure 5 which is fundamental to the conclusion, we find that the same analysis as above gives T-ARBS enrichment in SH with comparably low p-value $< 2e^{-16}$, which also holds true for PS/UN enrichment for Met-ARBS. Therefore, we conclude that although these cutoffs have no strict statistical foundation, the results flowing from these cut-offs are robust and do not change interpretation nor conclusion when altered. We have now included these points and description of these findings in the results and discussion sections.

2. Line 81 and Fig1b and c, it seems that there are significantly difference between the SH ARBS between the LNCaPBR, 22Rv cells vs the LNCaP and VCaP cells, which may suggest those SH ARBS contribute to ADT resistance? what about PS and UN ARBS between those lines? authors should also test some other enzalutamide resistant LNCaP sub lines such as LNCaP-95 or C4-2.

The reviewer has made an intriguing observation which prompted us to expand our analysis in these figures. Indeed, enzalutamide resistant LNCaP subline 42D^{ENZ^R} has an equally SH-ARBS enriched cistrome as the bicalutamide-resistant LNCaP derivatives. To provide a clearer overview of these data, we have now re-ordered the PCa cell lines based on the PCa stage that these models reflect (primary > metastatic > resistant) and observe a shift towards SH ARBS in acquisition of therapy resistance. These data are now included in the revised manuscript Figure 1B,C, and described further in the results section.

However, as was also pointed out to reviewer #2, the clinically observed heterogeneity of ARBS cannot currently be phenocopied with the limited prostate cancer cell lines available to the field. Interestingly, in the clinical data we observed that the most-heterogeneous ARBS are associated with resistance to AR-targeted therapeutics and disease progression. We have expanded the discussion to highlight this nuance more and we explain the intrinsic difference between clinical observations versus some of the cell line based observations.

3. same Fig1c, is there significant more SH sites in MDAMB453 compared to LSHAR? looks like there is a clear difference, what would be the interpretation?

We thank the reviewer for highlighting this unclarity. The molecular apocrine breast cancer MDAMB453 was included in Figure 1C as an AR+ non-prostate cell line control. Now, we expanded the analyses for the data represented in this figure, making it more clear that the SH enrichment statistic in Figure 1C does not show enrichment of SH-ARBS, mainly since there are so few overlapping ARBS found in MDAMB453. To better highlight this, we updated Figure 1B with percentage bars that clarify better which fraction of primary ARBS are found in cell line models. Thus, with hardly any overlap of ARBS between the AR+ breast cancer cell line versus prostate cancer profiles, the low representation of peaks may suggest enrichment of SH-sites, but with hardly any regions that support such a conclusion. We mention this now in the results section.

4. Fig2e, it seems that mutation rate of SNVsn is decreasing across all ARBS from primary to metastatic samples? isn't this controversial to the accumulation of SNVs during PCa progression?

The reviewer is correct to state that such a decrease in mutation rate during PCa progression would be controversial, and our data are not intended to communicate this message. However, in our metric we had already corrected for the expected base mutational rate reported for primary PCa and metastatic prostate cancer, which is now better reflected in the updated methods and in axis labels of Figure 2E and its caption.

5. Fig3c, for all of the 25 essential genes, is there difference between their association with PS or UN sites? for example, UCHL5 interacts with significant number of PS sites, while ZBTB10 interacts with UN sites much more. what would be the interpretation of those differences between the 25 genes?

The reviewer raises an interesting point, yet it is hard to address since the visualized data in Figure 3C are gene-centric, with gene knockout data as reported in the Depmap CRISPR KO database (Ref. 40 Tsherniak et al., 2017 Cell), while the vast majority of the quantitative interpretations in the present manuscript are enhancer-centric. For instance, the ZBTB10 promoter interacts with a total of 32 ARBS (of which 2 SH, 20 PS and 10 UN), while the UCHL5 promoter interacts with 9 ARBS (of which 2 SH, 5 PS and 2 UN).

Making a correct interpretation is further complicated given that different genes have different number of enhancers acting at different levels of redundancy and genomic distances, making cross-comparison between genes complex. Instead, here we intend to highlight that many of these essential gene promoters are interacting with heterogeneous ARBS (PS- and UN-ARBS, >60% are found in 20 patients or less) as is presented in Figure 1D. As such, this limitation in interpreting this data further based on differences in association with PS- and UN-ARBS is now mentioned explicitly in the discussion section.

6. Fig4e shows the guides pair 12 is the most effective ones to inhibit CITED2, but the effect of other guide pairs is not shown (only the pooled one is shown). What about the other guide RNAs targeting the UN ARBS? Also, since these pairs of guide RNAs targeting the whole sites, it is necessary to perform rescue experiment to confirm the biological consequence is real due to the deletion of these sites.

We thank the reviewer for highlighting this point. Indeed, pair34 was not included in these analyses since not all guides were effective within a pair, resulting in non-detectable KOs on genomic DNA level (Supplementary Figure 6D). Since the ineffectiveness of a single guide in a pair may cause this, we combined all guides in a pool to maximize the chance of a KO. Illustrated below are the other guide RNAs targeting UN-ARBS CI1 and PS-ARBS CI8 in which this effect is more clearly visible.

We agree with the reviewer that additional experiments are required to confirm that the biological consequence of enhancer perturbation is real. Therefore, we have developed and implemented a robust CRISPR inhibition system based on Suntag (Ref. 43 Tanenbaum et al., 2014 Cell), which recruits up to 10 Krüppel Associated Box (KRAB) domains for effective repression of genomic loci. Additionally, we designed new guides that are targeted much more closely to AR motifs for more robust repression of putative enhancers at loci inside these regions rather than targeting enhancer flanks for element removal (Supplementary Figure 6A,C). The results of this orthogonal method, in which we do not remove whole sites, but instead epigenetically suppress their activity, are shown in the new Supplementary Figure 6F and are in agreement with our site deletion transcriptomic experiments. Importantly, these results support the original conclusion that less-often shared ARBS can have the greatest impact on transcriptional output.

7. Fig4F, G, what are the gene motifs enriched for AR13 sites beside HOXB13? Do HOXB13 motifs also enriched in other AR2 or AR23 sites? more importantly, the author should show there is indeed downregulation of HOXB13 signature in those AR13 KD cells, and perform rescue experiment rescue the cell proliferation.

These points are very interesting , which we have thoroughly investigated. First, we have included PCa-relevant motif analysis on the AR-interacting ARBS in Supplementary Figure 7B, which includes motifs such as AR, FOXA1, HOXB13 and GATA2 and ETS1. Importantly, this supplement shows that we observe exclusive HOXB13 motif enrichment for site AR13, which is recapitulated by Supplementary Figure 7C, which shows HOXB13 binding at these loci from publicly available HOXB13 ChIP-seq data for LNCaP and 22Rv1 (Ref. 15 Kron et al., 2017 Nat. Genet.).

The reviewer raises a second interesting point regarding HOXB13 binding in AR13 KD cells, which were not included in our first submission. With CRISPRi Suntag-KRAB LNCaP with guides targeted at AR13, we first confirmed the impact of AR13 suppression on proliferation (Supplementary Figure 7D), which are in agreement with the previously reported data of Ref. 13 Takeda et al. 2018 as presented in Figure 4G. Rescue experiments of enhancer activity would not be feasible, as eRNAs often act locally at the genomic locus of origins (Kim et al. 2010, Nature; De Santa et al., 2010 PLOS Biol.) and exogenously introduced eRNAs would as a consequence not serve as proper rescue-experiments for the phenotype. This is now better explained in the discussion section.

Finally, to validate the functional involvement of AR13 in chromatin binding of HOXB13 at this region, we performed HOXB13 ChIP on Suntag-KRAB LNCaP with either NT or AR13 sgRNAs. These data show that HOXB13 binds to AR13, which was decreased when AR13 was targeted by CRISPRi, thereby showing that perturbation of AR13 directly leads to a decrease in HOXB13 binding compared to NT (Supplementary

Figure 7E) and a concomitant drop in proliferation capacity (Supplementary Figure 7D). Taken together, these data support our hypothesis that AR13 enhancer action involves HOXB13 and is unique compared to the other ARBS in this regard with strong evidence.

8. Fig4F, G, the authors didn't explain why AR13, 23 was chosen here, vs the ARBS shared by 54 patients in previous analysis? what is the rationale of choosing all different cutoffs of shared ARBS throughout the paper?

We thank the reviewer for these questions and apologize for the confusion. We would like to clarify that AR13 and AR23 (interacting with the AR promoter on ChrX) concern different enhancers than CI54, which interacts with the CITED2 promoter on chr6. Along these lines, each AR-regulated gene has different numbers of enhancers, that are shared among different tumors to variable degrees. We have updated the text in the results section to highlight this issue better, and we sincerely hope this additional information suffices to resolve the confusion.

9. Fig5B, how many poor vs good outcome ARBS are SH vs PS vs UN sites? Are there enrichment of poor outcome ARBS for the UN sites compared to SH sites?

SH-ARBS overlap with 60 good and 48 poor outcome sites, while PS-ARBS overlap with 78 good and 149 poor outcome sites. There are no UN-ARBS in poor vs good outcome ARBS, which is likely related to the lower quality of AR CHIP-seq data from the original dataset (Ref. 8 Stelloo et al., 2015 EMBO Mol. Med.). When doing Fisher's exact test, we find an enrichment of SH-ARBS in both good and poor outcome sites ($p < 2.2E-16$). In contrast, PS-ARBS only show enrichment in poor outcome sites with $p = 0.002117$.

For more clarity on these data, we have additionally provided an overview of these sites in Supplementary Table 5 and 16, which is now also graphically represented by Supplementary Figure 8B.

10. Fig5D, what about the met vs primary ARBS in the SH ARBS? no difference? is there enrichment of those met-ARBS, especially the ones at UN-ARBS, enriched in any genes known to promotes PCa metastasis?

We thank the reviewer for raising this interesting point. Per reviewer request, we have now recalculated these and find no enrichment of either primary sites or met-ARBS in SH ARBS with respectively p value = 1 (just a 1 primary ARBS found in SH-ARBS) and p value = 1 (61 met-ARBS found in SH-ARBS).

To address the second question, we have analyzed the heterogeneous case-specific met-ARBS (now also presented in Supplementary Table 18) more closely for PCa metastasis promoting genes rather than only genesets or pathways. Next to lipid biosynthesis and cholesterol homeostasis genes APOC3, SLC25A2 and OSBPL8 we also find proto-oncogene RET which plays a role in cell differentiation, growth, migration and survival. Other potential migration related genes regulated by heterogeneous case met-ARBS are cell-cell adhesion genes CDH17,18 and ITGB5,7, while FOS2L is an important transcription factor controlling proliferation and transformation of cells, especially osteoclasts, of which activation plays an important role in the development of bone metastases. The genes that are known to promote PCa metastasis that overlap with heterogeneous met-ARBS have been included in Supplementary Table 18. Interestingly, also an ARBS regulating tumor suppressor gene CDKN2A is found, which is likely additionally affected by SVs or mutations in metastatic disease.

11. For the STARR-seq data, it's better to validate specific inducible/constitutive/inactive STARR-seq enhancers in vitro using luciferase reporter assay.

Yes, we fully agree. Following the reviewer's advice, we included additional luciferase reporter assays to validate the STARR-seq results. For this, we've selected 14 regions that were called inducible/constitutive/inactive in the STARR-seq results and cloned these in luciferase constructs. Luciferase assays on LNCaP cells in EtOH/DHT conditions confirmed the specific behavior of these ARBS as shown by STARR-seq in the newly added Supplementary Figure 2C,D, which we now also describe in the results section.

Reviewer #4 (Remarks to the Author): Expert in prostate cancer epigenomics

Kneppers et al performed analyses using publicly available data (generated from Zwart lab + other researchers) to study androgen receptor binding sites in primary prostate cancer patients. Many useful integrative analyses are done, but it is not clear what is the key take-home/novel message of this manuscript. Researchers already know that there are heterogeneous epigenetic signatures among prostate tumor samples. See below to find specific comments/questions of each analysis/figure.

We sincerely thank the reviewer for the critical and constructive comments and express our gratitude to the kind comment on the utility of the integrative analyses. Indeed, the field has awareness of epigenetic heterogeneity in prostate cancer, which we fully acknowledge, but to our knowledge to prior studies have analyzed this inter-tumor epigenetic heterogeneity in depth. As such, we believe the current study brings key novel insights on the genetic basis and biological consequences of this epigenetic heterogeneity among tumors. As main take-home messages, the manuscript reports:

1. AR enhancer usage is highly heterogeneous between tumors. This is relevant since most prior reports in the field have used consensus-lists of AR sites between tumors, effectively ignoring those sites that are not shared by less than half of the patients and focusing only on commonalities.
2. This inter-tumor heterogeneity of AR action is poorly represented by cell line models that are most-commonly used in the field.
3. Most-commonly shared ARBS are most-active and enriched for SNPs and somatic mutations, but large-scale structural variation affects both commonly shared and least-commonly shared ARBS.
4. The least-commonly shared ARBS are clinically informative, showing most overlap with metastasis-defining ARBS, and are selectively enriched in tumors from patients who develop a biochemical relapse later in life.

We sincerely hope the reviewer agrees with these take-home messages as being novel and informative beyond the current state-of-the-art and considers the work of sufficient relevance for publication. We have amended the manuscript to convey these take-home messages more clearly. Regarding all other issues that were raised by the reviewer, please see below in our point-by-point response. With this, we sincerely hope the reviewer finds the manuscript sufficiently improved, and suitable for publication.

Figure 1: Averaging 5,787 peaks per tumor seems to be too low and the number of identified peaks per tumor seems to vary a lot. Detailed information of identified peaks per tumor is needed as technical

variation can lead to those variations. It is not clear if the author evaluated quality of each dataset (e.g. use ENCODE guideline to measure NSC, RSC to check enrichment level, read numbers) besides checking GC contents. Provide how many peaks are found from each sample, how many are categorized to SH/PS/UN per tumor. Current suppl table/figures do not include those.

The reviewer raises highly important points related to reproducibility, technical variation and QC of the peaks identified in these samples. Standard QC measures (NSC, RSC, read numbers, FRiP score) were included in the (Ref. 3 Stelloo et al., 2018 Nat. Commun.), and have previously been rigorously evaluated for these quality parameters. To enhance clarity and transparency, these parameters are now included again as Supplementary Table 4 in the current study. Furthermore, we would like to mention that AR peak numbers vary greatly between samples, but based on our extensive analyses and QC, we conclude that this variation has a biological rather than a technical nature, highlighting once more the high level of inter-tumor epigenetic heterogeneity. As additional QC steps beyond sequencing-based measures, we have now included below additional information on AR expression level on IHC for the primary tumor/healthy tissue comparisons for a subset of samples, illustrating that the vast majority of cells are AR positive.

Subject	Tumor enrichment in ChIP-seq sample (%)	AR		Gleason Score
		Normal prostate (%)	Prostate tumor (%)	
NKI_1	60	90	80	4+5=9
NKI_5	65	80	90	3+4=7
NKI_7	70	90	90	3+4=7
NKI_13	30	90	90	3+4=7
NKI_20	70	90	70	3+4=7
NKI_23	40	80	90	3+4=7
NKI_27	30	90	90	4+3=7
NKI_29	40	90	80	4+4=8

Furthermore, following the reviewers recommendation, we now provide the SH-PS-UN distribution for each sample separately, in which 7,394 peaks are found on average per tumor tissue sample (Supplementary Figure 1E and Supplementary Table 3) instead of the previous erroneously reported number 5,787. We thank the reviewer for highlighting this typographic mistake, and we have updated the main text with the correct numbers and supplementary tables. Of note, these numbers are in the same or a higher order of magnitude as compared to other AR ChIP-seq datasets reported in primary tumor samples published over the years (Ref. 4 Pomerantz et al., 2015 Nat. Genet. 2015, Ref. 5 Pomerantz et al., 2020 Nat Genet. and greatly outnumbering Ref 8. Stelloo et al., 2015 EMBO Mol Med). As we reported previously, low number of AR peaks in specific samples are associated with decreased AR activity in these samples -while AR levels were comparable to other samples in the cohort (Ref. 3 Stelloo et al., 2018 Nat. Commun.), and represent a primary tumor subtype that is hallmarked by low AR activity yet potentially driven by other pathways.

These data jointly support our previously-reported conclusion that variation in the number of AR peaks between samples are biologically meaningful. In all fairness, we would like to note that not all samples pass ENCODE QC parameters, especially on the fraction-of-reads-in -peaks, but as ENCODE criteria are based on cell lines, we respectfully are not convinced this is a fair comparison. However, we have analyzed all publicly available ChIP-seq data in cell lines used in this study, based on ENCODE QC parameters and

have found that these all comply to 6/6 ENCODE QC parameters, which has been included in Supplementary Table 1.

Describe whether peak height (ChIP-seq signal) is considered to rank peaks besides overlapping identified peaks among datasets (SH/PS/UN). False positive peaks may exist in unique ones.

We thank the reviewer for suggesting this excellent quality control measure. Following the reviewer's advice, we have now performed these analyses by correlating peak height with SH-PS-UN bin for each patient. As expected from the trends observed in the strength of the ChIP-seq signals of SH/PS/UN ARBS groups as presented in Figure 1E,F, we find an inverse correlation between SH-PS-UN bins, which is visually represented in the new Supplementary Figures 4B,C.

Figure 1D and reference 7: Authors stated that AR enhancer heterogeneity is not tumor-intrinsic, but instead patient-intrinsic. Generate Figure 1D-like figure using cancer-specific/normal-specific ARBS only to test that. Explain how patient-intrinsic ARBS can lead to transcriptome changes linked to cancer subtype/metastatic cancer.

We apologize for this unclarity. While we observe strong inter-patient variations in AR peaks, as highlighted in Figure 1D, reproducible patterns of specific subsets of ARBS are observed when comparing groups, as we previously reported for the 100 primary prostate cancer cohort (Ref. 3 Stelloo et al., 2018 Nat. Commun.), but also when comparing healthy prostate epithelial tissue with primary tumors (Ref. 34 Mazrooei et al., 2019 Cancer Cell and Ref. 5 Pomerantz et al., 2020 Nat. Genet), but also between primary tumors versus mCRPC PDX samples (Ref. 5). Along those lines, the specific subsets of ARBS that robustly stratify tumors on specific tissue states or subtypes, are also associated with differential gene expression programs that separate these samples. We have updated the discussion section to explain these results better.

As suggested by the reviewer, we further investigated the overlap of NARBS and TARBS (Ref 4. Pomerantz et al., 2015 Nat. Genet.) in normal and tumor ARBS rankings, which shows a similar trend (Supplementary Figure 1F). These new analyses support the conclusion that heterogeneity is indeed patient-intrinsic, validating our original conclusion.

We hypothesized that the diversity of active ARBS that are already detectable in primary PCa patient may later regulate a variety of transcriptomic changes that underlie metastatic phenotypes. Indeed, the data in Figure 5 shows that these patient-intrinsic ARBS have the potential to drive metastatic transcriptomic changes that are mostly found in patients with a considerably worse prognosis.

Figure 1H, Suppl Fig 1: Motif analysis - Authors indicated that MED1 and RNAPII subunits were enriched in UN-ARBS. How many peaks have each motif? Are these located in promoter/enhancer? Enrichment score is not informative if the number of peaks with motifs is too small. Provide more information and interpret data more thoroughly.

We apologize for the confusion. The analysis shown in Supplementary Figure 1 does not depict motifs, but enrichment of experimentally confirmed overlap for the factor highlighted using GIGGLE (Ref. 64 Kumar et al., Nat. Biotechnol. 2013). GIGGLE is a tool to comprehensively search many public epigenome experiments for genomic locations of histone modifications, TFs and other core transcriptional regulatory components like MED1 and RNAPII. Although the latter two proteins bind the DNA, these do not function as transcription factors and therefore do not have specific motifs.

However, as per the reviewer's request we provide more information regarding the genomic location in Supplementary Figure 1J which shows genomic element locations that are typically associated with MED1 and RNAPII (both promoter and enhancer). Indeed, as this figure shows, the scores are no longer informative for SH-ARBS compared to PS and UN, as SH-ARBS concern a relatively small number of peaks. We have also updated the results section to mention this limitation.

Figure 1H: Letters are too small and it's not informative. Either replace with a barplot or move to supplemental figures.

We thank the reviewer for the suggestion and have replaced the three wordclouds with a more informative barplot detailing z-scores for the main TF families identified in Figure 1H, thereby facilitating direct comparison between SH-, PS- and UN-ARBS.

Figure 2A: Not clear if machine learning is appropriate to analyze patient AR ChIP-seq data for a massive parallel reporter assay data generated in one cell line. As Figure 1 showed, the number of ARBS and ARBS differ among patients.

We fully agree with the reviewer that the number of ARBS differ among patients and that this could influence the machine learning models. However, while this would impact the identification of individual ARBS enhancers, we have demonstrated that these machine learning models can robustly identify active enhancers as a group. Specifically, we observed a strong correlation of H3K27Ac signal in clinical studies (Ref. 63 Linder et al., 2022 Cancer Discovery) to the STARR-seq results in LNCaP. Similar to other enhancer associated features (e.g. RNA polII, eRNA, histone marks) these predictive models can therefore be thought to broadly correlate with clinical activity. We have updated the discussion section to reflect this nuance.

Authors performed additional STARR-seq targeting at 2,495 randomly sampled heterogeneous ARBS, which is good. However, more information needs to be provided. How many were found in shared/PS/unique and how many are found in ind/cons/inactive enhancers? Provide those in suppl data. What is the take-home message? Unique ones are less functional?

We are happy to see the reviewer appreciates the additional STARR-seq experiments. Following the reviewer's recommendation, we now provide the requested information about the 2,495 randomly sampled ARBS in the new Supplement Table 8 and visualized these in the new Supplementary Figure 2A,B. This data indicates that some of the UN-ARBS are active enhancer elements in a STARR-seq LNCaP context, conveying the message that UN-ARBS can possess intrinsic enhancer functionality. Finally, we have updated the results section to better reflect this message.

Figure 2D-F: Are rSNP/SNV alleles from primary AR ChIP-seq data used to evaluate rSNP/SNV allele or just genomic location of reported rSNP/SNV regions were overlapped to ARBS? It would have been better if authors could have done more analyses using allele information from ChIP-seq data.

We thank the reviewer for this excellent suggestion. While in our initial analyses we have not included allelic imbalance analyses from AR ChIP-seq data, following the reviewer's advice, we have now included these analyses in Figure 2D,E. Using a previously reported Cistrome Wide Association Study (CWAS) pipeline on our data (Ref. 30 Baca et al., 2022 BioRxiv ; in press Nat. Genet.), we were able to impute germline SNV allelic imbalance data from primary patient ChIP-seq with FDR q-value < 0.05 and overlap

these with our ranked ARBS, showing an enrichment of these imputed primary cQTL in SH-ARBS. This is now also reflected in the results section.

Figure 3A: Promoter-enhancer interaction data used from this study is from one cell line while AR ChIP-seq data is from many patients (primary tumor samples).

The reviewer is correct in their assessment that interaction data is solely from LNCaP, but this dataset has been used recently to infer promoter-enhancer interactions from other patient series in prior reports from us and others (Ref. 38 Giambartolomei et al., *Am J Hum Genet.* 2021; Ref. 5 Pomerantz et al., 2020 *Nat. Genet.*; Ref. 30 Baca et al., 2022 *BioRxiv*, in press *Nat. Genet.*). Moreover, these data are in line with previously reported VCaP AR ChIA-PET data (Ref. 39 Zhang et al., *Genome Res.* 2019). Nonetheless, we do agree with the reviewer that additional promoter-enhancer interaction datasets would be valuable to further support these conclusions through other independent means. Therefore, we included single cell ATAC-seq data (Ref. 14 Taavitsainen et al., 2021 *Nat. Commun.*) in Supplementary Figure 5, which shows that the single cell ATAC-seq data from Ref. 14 is in agreement with the H3K27ac Hi-ChIP data regarding promoter-enhancer interactions through computed co-accessibility links (Ref. 85 Pliner et al., 2018 *Mol. Cell*). Prior to this, ATAC-seq has been applied to infer promoter-enhancer interactions in Corces et al., 2018 *Science*.

Figure 3B: Are large CNA gains/losses tested in AR ChIP-seq data? Just genomic location of reported CNA regions were overlapped to ARBS? It would have been better if authors could have done checked CNA using ChIP-seq/RNA-seq data and show CNA relationship to ARBS using the datasets from the same sample.

We thank the reviewer for highlighting this issue. These data are available through supplemental data of Ref. 3 Stelloo et al., *Nat. Commun.* 2018, which has inferred CNAs of primary tissue from AR-ChIP-seq data through CopywriteR (Kuilman et al., 2015 *Genome Biology*, <https://github.com/PeeperLab/CopywriteR>). In the primary disease stage, large CNA gains/losses that emerge during metastatic disease are not yet present. Since we were interested to elucidate whether such ARBS could later play a role in gene regulation, we used metastatic tumor sequencing data from a highly cited resource (Ref. 11 Quigley et al., 2018 *Cell*) and inquired whether large scale genomic regions containing prostate cancer-dependent genes and their interacting regulatory elements would be affected (Figure 3). Indeed, CNAs are ubiquitously present in metastatic disease, but we show in Figure 3E that such CNAs at exclusively enhancer elements have similar transcriptional outcomes to CNAs at the coding sequence (using matched metastatic RNA-seq data). Taken together, these data suggest that heterogeneous ARBS in primary patients are affected by CNAs in metastatic disease which leads to equally modified transcriptional response of prostate cancer dependent genes.

Figure 4 and Suppl Figure 4: Figure 4C and D are done in what cell line? Both LNCaP and 22Rv1 or just one cell line? Are CRISPR/Cas9 experiments done in pool or single cells are cloned and performed RT-qPCR? It looks like that it's in the pool, right? If that's the case, H3K27ac ChIP is done to check the deletion effect in the pool? What controls are included? Explain more in detail.

We thank the reviewer for flagging up this unclarity. Following the reviewer's advice, we have now clarified these points in the main text and methods, and explained further below. We present an updated Cas9 western blot to show LNCaP and MDA-MB453 Cas9 expression, both of which were used in the study

(Supplementary Figure 6A). CRISPR/Cas9 experiments were performed in polyclonally puromycin-selected cells, but no new H3K27ac ChIP has been performed as deletions were assayed by gDNA locus specific amplification and RT-qPCR, with a NT control guide pool and two different exon-exon junction spanning qPCR pairs, which were also combined to increase successful KO chance in case of an inefficient sgRNA.

However, to show these effects without introduction of double-strand breaks, we have developed a Suntag-based (Ref. 43 Tanenbaum et al., 2014 Cell) CRISPRi system that strongly represses enhancer sequences without deletion effects through recruitment of up to 10 Krüppel Associated Box (KRAB) domains. We validated Suntag-KRAB expression on western blot and flow cytometry and generated monoclonal cell lines. Additionally, we confirmed its function by perturbing expression of EPCAM, which is highly expressed by LNCaP, and found that anti EPCAM-APC staining on flow cytometry shows a marked decrease in EPCAM expression in EPCAM targeting versus NT sgRNAs. More importantly, this orthogonal method shows comparable CITED2 enhancer perturbation results on RT-qPCR as compared with CRISPR/Cas9 knockouts using classical Cas9-eGFP, both suggesting that the most-commonly shared ARBS is not necessarily the most-influential ARBS for transcriptional activity, as is shown in Supplementary Figure 6C,F.

Figure 5. TARBS and met-ARBS analysis finding (e.g. lines 418-422) is contradictory to the recent manuscript ref 7? Explain. It is not clear how transcription levels of ARBS' target genes are studied. Are HiChIP data used to identify target genes or nearby genes are used? Explain. Provide a list of identified genes in suppl table.

The reviewer lists a number of relevant issues. We would like to highlight that the work in Ref. 7 Severson et al., 2021 Mol. Oncol. was performed on clonal metastases derived from a single patient autopsy. Since >80% of peaks overlapped between samples, reaching a level seen in technical replicates, samples can be considered homogeneous. In contrast, our ARBS ranking was obtained from many primary prostate cancer patients and exhibits high levels of epigenetic heterogeneity.

Indeed, as the reviewer rightfully highlights, Ref. 7 describes a set of ARBS shared among all 4 metastatic lesions from this patient, which showed strong overlap with peaks observed in normal prostate tissue and primary tumors (Ref. 7, Figure 3). However, this analysis was performed on all peaks shared between these metastases from within a single patient. This analysis in Ref. 7 was however not performed on metastasis-unique peaks exclusively observed in these metastases, and absent in primary tumors. Importantly, this specific analysis was performed by us in Ref. 5 Pomerantz et al., 2020 Nat. Genet., identifying those ARBS that are shared among all primary tumors (primary ARBS), versus those that are selectively gained in metastases (Met-ARBS). Our current analysis illustrates that the gained ARBS in metastases (Met-ARBS) are already pre-existing as highly-heterogeneous ARBS (detected only in <10% patients) that present selectively in primary tumors from patients who progress later in life as defined by biochemical recurrence.

With that, we conclude that these data are not contradicting our previous report, since in the current manuscript we focus specifically on those newly acquired ARBS in metastatic lesions. We have extended the discussion to make this key difference more clear.

We have used the GO Regulatory Potential metric as described in the methods, which not only uses distance between promoter-enhancer pairs but also adjusts these scores and ranks elements based on integration of ChIP-seq and expression data to identify target genes (Supplementary Figure 8C), which we

have now described better in the results section. Finally, as per the reviewer's request, a list of these identified genes has now been added to the new Supplementary Table 18, including a list of known metastasis-driving genes that interact with heterogeneous met-ARBS in exclusively patients with poor prognosis. These genes are involved in biosynthesis and cholesterol homeostasis (APOC3, SLC25A2 and OSBPL8), or are proto-oncogenes such as RET which plays a role in cell differentiation, growth, migration and survival. Other genes regulated by heterogeneous case met-ARBS are involved in migration and invasiveness, such as cell-cell adhesion genes CDH17,18 and ITGB5,7, while FOS2L is an important transcription factor controlling proliferation and transformation of cells, especially osteoclasts, of which activation plays an important role in the development of bone metastases.

REVIEWER COMMENTS

Reviewer #1 (Remarks to the Author):

Authors have done excellent work in addressing the issues raised in my initial review. I am happy with their revision and have no more concerns with the work.

Reviewer #2 (Remarks to the Author):

My comments have been addressed.

Reviewer #3 (Remarks to the Author):

The authors did a fantastic job addressing questions and concerns. The manuscript is significantly improved and I have no further concerns.

Reviewer #4 (Remarks to the Author):

After seeing the responses from authors, I have become suspicious about ARBS the authors used for downstream analyses. New Table S4 now includes QC stats of AR ChIP-seq data that includes NSC, RSC, and FRiP score. A lot of AR ChIP-seq data did not pass the NSC, RSC, and FRiP scores ENCODE ChIP-seq guidelines set. Actually, by sorting datasets by NSC/RSC/FRiP score, we can know that there are lots of AR ChIP-seq data which resulted in smaller number of peaks have bad quality (low NSC, low RSC, low FRiP). Including bad quality datasets will include many false positive peaks, leading to the wrong direction. Screenshots of genome browser make me worried about ARBS they selected and used for analyses too; Specifically, many UN ARBS seem to be false positives. As these datasets are major and big foundation resources for analyses performed in this manuscript, I am concerned about findings and conclusions authors made. Bioinformatic analyses do not seem to be informative and novel as they are performed mainly using datasets from cell lines, which authors claimed and agreed that they are very heterogeneous from tissues'. Moreover, I am still confused what are take-home messages and novel findings of this manuscript.

After seeing the responses from authors, I have become suspicious about ARBS the authors used for downstream analyses. New Table S4 now includes QC stats of AR ChIP-seq data that includes NSC, RSC, and FRiP score. A lot of AR ChIP-seq data did not pass the NSC, RSC, and FRiP scores ENCODE ChIP-seq guidelines set.

We thank the reviewer for the constructive feedback and critical suggestions.

The hypothesis of our work is that while such consensus building generally results in highly robust binding site universes describing commonalities between samples, the process of consensus building inevitably entails loss of rare true positive binding sites specific to a minority subset of patients. The rationale of our methodology was thus to not exclude any samples *a priori*, in order to prevent any form of bias or consensus building.

As the reviewer correctly indicates, ChIP-seq experiments are standardized according to ENCODE guidelines, but these guidelines are designed based on cell line observations. Tissue ChIP-seq is technically more challenging and usually suffers from lower quality metrics, due to tissue collection, inherent tissue heterogeneity and processing prior to ChIP-seq. The resulting sequences are also processed according to ENCODE guidelines through standardized computational pipelines, a process in which sequencing data is cleaned, aligned and ChIP-seq peaks are called using MACS. Finally, analysis is performed, a process in which the field often employs the practice of consensus building.

However, the reviewer is justified in flagging up concerns regarding ChIP-seq quality of some of the tissue samples, especially since these are foundational to many of the analyses in the present manuscript. We have therefore carefully re-examined lower quality samples and have attempted to replicate critical analyses when excluding such samples. The current ENCODE ChIP-seq standards (ENCODE4, July 2020) are publically available through <https://www.encodeproject.org/chip-seq/transcription-factor-encode4/#standards>.

Specifically, there are target-specific standards:

“Each replicate should have 20 million usable fragments.

low read depth: 10 million to 20 million usable fragments

insufficient read depth: 5 million to 10 million usable fragments

extremely low read depth: < 5 million usable fragments”

We have now created the new Supplementary Figure 2, in which panel A shows which samples comply to which category based on all single mapped reads. In summary, 6/88 samples have a read depth (RD) considered by ENCODE guidelines as insufficient, while the RD of a vast majority of the samples range from the high end of low RD to high RD. No association of low RD with other QC parameters were observed (NSC, RSC, FRiP), as now included in Supplementary Figure 2A,C.

Additionally, the most-recent ENCODE standards mention the following:

“Additional metrics are calculated without defined thresholds, such as fraction of reads in peaks (FRiP), and are useful in comparing similar experiments.”

The ChIP-seq data used as foundations for this work, were reported by us previously (Stelloo et al., 2018 *Nat. Commun.*) along with the QC metrics, as we referred to in the first submission. In this previous report, we identified a subpopulation of tumors with low AR chromatin binding (thus low FRiP scores) and low AR activity (based on RNA-seq) in which other pathways -including FGFR signaling- may be driving the tumor instead. In other words: the low number of peaks for AR in these specific samples appeared to be biological, not technical, and removing these samples beforehand would have pushed the biological interpretation of our data into a specific and biologically biased direction.

When comparing the used dataset to other publically available tissue AR ChIP-seq experiments, the metrics suggest that the dataset we used as a foundation can be considered among those of the highest quality clinical sample ChIP-seq series. This is reflected by the widespread foundational use of this resource by the field, which besides the studies already referenced in this manuscript (Ref. 5 Pomerantz et al. 2020 *Nat Genet.*, Ref 12. Huang et al. 2021 *Genome Biol.*) additionally includes large multicenter studies such as Houlahan, Shiah et al., 2019 *Nat Med.* and Zhao et al., 2020 *Nat Genet.*, but also studies that focus on specific prostate cancer mechanisms (Li, Yuan, Di, Xia, et al., 2020 *J. Clin. Invest.*; Grbesa et al., 2021 *Cell Rep.*; Rodríguez et al., 2022, *Cancer Res.*). Finally, we would like to emphasize that tissue transcription factor ChIP-seq typically is characterized by lower additional metrics such as NSC, RSC and FRiP score.

Actually, by sorting datasets by NSC/RSC/FRiP score, we can know that there are lots of AR ChIP-seq data which resulted in smaller number of peaks have bad quality (low NSC, low RSC, low FRiP). Including bad quality datasets will include many false positive peaks, leading to the wrong direction. Screenshots of genome browser make me worried

about ARBS they selected and used for analyses too; Specifically, many UN ARBS seem to be false positives. As these datasets are major and big foundation resources for analyses performed in this manuscript, I am concerned about findings and conclusions authors made.

To formally address the reviewer's concerns whether sample quality could indeed influence the direction of the major conclusions of this work, we have now included a full re-analysis by using ENCODE read depth guidelines among a continuum of exclusion criteria groups to see how excluding lower quality samples would affect ARBS heterogeneity (Supplementary Figure 2B,E). We observe that the findings presented in Figure 1 are robust under these exclusion criteria. However, highly stringent exclusion criteria leads to a cohort of 53/88 patients, at the potential cost of losing ARBS heterogeneity and the biased preselection of tumors with particular biological features, as explained above. To make this insightful, we traced the origin of UN-ARBS in the context of sample quality and observed equal density distributions of UN-ARBS coming from different RD quality samples across the genome as compared to all originally reported UN-ARBS (Supplementary Figure 2D).

Out of all samples, P394T is likely of the lowest quality (insufficient RD, low FRiP score) which affected the distribution of SH/PS/UN-ARBS compared to all other samples, with UN-ARBS making up roughly half of the ARBS detected in this patient (Supplementary Figure 1E). To investigate whether it is prudent to either include or exclude this sample in the context of understanding ARBS heterogeneity, we carefully re-examined MSPC calculated scores and found that 54/71 of P394T's UN-ARBS pass statistical testing ($-\log(p\text{-value}) > 8$, Supplementary Figure 2F). Moreover, MISP motif screening analyses detected AR motifs in these UN-ARBS, but more importantly also in those that did not pass MSPC statistical testing (Supplementary Figure 2G). To investigate whether RD would affect the distribution of MISP AR motif screen scores in UN-ARBS, we then ran MISP on all UN-ARBS and those originating from either high, low or insufficient RD samples. Besides a limited increase of low or undetectable AR motifs in samples with insufficient RD, we observed comparable MISP motif score distributions (Supplementary Figure 2I). Most importantly, samples classified as low RD contributed similarly to AR score distributions compared to those in high RD.

Cumulatively, these analyses underline that the impact of sample quality on direction or conclusions drawn in this study is limited. Moreover, a majority of UN-ARBS are in fact detected in high RD samples and examination of lower quality samples suggests that further exclusion these samples on top of existing stringent quality control measures would lead to biases in

mapping primary prostate cancer ARBS heterogeneity. This has been further confirmed by AR motif screening, which shows similar AR motif score distributions in UN-ARBS across sample quality (Supplementary Figure 2I), but also earlier by the experimental verification of enhancer activity at some of these UN-ARBS through STARR-seq (Figure 2B,C and Supplementary Figure 3). These findings suggest that patient's cistromes are also constituted by binding events that are rare inside the total population.

Bioinformatic analyses do not seem to be informative and novel as they are performed mainly using datasets from cell lines, which authors claimed and agreed that they are very heterogeneous from tissues'.

We would like to highlight that cell line data in our manuscript are mainly limited to functional biological experiments and analyses, and computational analyses in cell lines only represent a minority of our conclusions. We respectfully disagree with the reviewer on the level of novelty in the cell line experiments, as no other report to date to our knowledge has analyzed the activity of AR binding sites in relation to the inter-patient heterogeneity thereof. This stresses both the novelty and informative nature of our findings, especially as patient-unique binding sites in ChIP-seq analyses are typically discarded in most studies. The main take-home messages of the paper, as discussed further below, are represented in 22 panels in 5 main figures, and are exclusively based on human tumor data.

Moreover, I am still confused what are take-home messages and novel findings of this manuscript.

The field's practice of building consensus cistromes from multiple tissues may inherently carry the risk of losing the capacity to infer epigenetic heterogeneity, despite its usefulness. Indeed, we and others have conducted many human sample cistromic studies using consensus building as methodology and the practice has led to new key insights in the development and progression of endocrine-related cancers (Yu et al., 2010 *Cancer Cell*; Sharma et al., 2013 *Cancer Cell*; Ref. 4 Pomerantz et al., 2015 *Nat. Genet.*; Chen et al., 2018 *Proc Natl Acad Sci U S A*; Patten, Corleone et al., 2018 *Nat Med.*; Chi, Singhal et al., 2019 *Proc Natl Acad Sci U S A*, Ref. 5 Pomerantz et al., 2020 *Nat. Genet.*; Ref. 6 Baca et al., 2021 *Nat. Commun.*). However, more personalized therapy would likely be beneficial to patient care, since primary PCa tumors at time of radical prostatectomy are frequently multifocal and additionally also heterogeneous between patients (Rubin, Demichelis 2018 *Modern Pathology*), suggesting that in this stage epigenetic heterogeneity is already present.

With this work, we show that inter-patient heterogeneity of AR binding sites in primary PCa patients bears biological information with the most-highly conserved AR sites between tumors enriched for somatic mutations and germline-SNPs and highest in enhancer activity. However, the least-conserved AR binding sites between patients bear clinically most-relevant information, with rare mCRPC-enriched UN-ARBS already to occur exclusively in primary tumors from patients destined to develop progressive disease later in life. This observation is novel, and highlights the clinically relevant, but highly understudied nature, of patient unique AR binding sites, which to date has not been appreciated by the field.

REVIEWERS' COMMENTS

Reviewer #4 (Remarks to the Author):

I appreciate the authors who performed additional analyses and generated Figure S2.

Reviewer #4 (Remarks to the Author):

I appreciate the authors who performed additional analyses and generated Figure S2.

We thank the reviewer for critically assessing data quality and comments, which resulted in higher transparency in data quality and improved manuscript quality.